# Nitrate-nitrogen dynamics in response to forestry harvesting and climate variability: Four years of UV nitrate sensor data in a shallow, gravel aquifer

Ben Wilkins[1], Tom Johns[1], Sarah Mager[2]

[1]Environment Canterbury Regional Council, Christchurch, 8011, New Zealand

[2]University of Otago, New Zealand, Dunedin, 9016, New Zealand

*Correspondence to*: Ben Wilkins (ben.wilkins@ecan.govt.nz)

**Abstract**

The leaching of inorganic nitrogen can adversely affect the quality of groundwater and its hydrologically connected streams and rivers. Traditionally, these effects have been assessed using discrete, low frequency water quality measurements. However, it is difficult to characterise the complex biogeochemical processes that control nitrate-nitrogen dynamics in groundwater when using temporally sparse data. In this study, we installed a high-frequency UV nitrate sensor, down gradient of plantation forestry in a shallow, gravel aquifer to understand nitrate-nitrogen dynamics in groundwater. We found that there were two mechanisms of nitrate-nitrogen pulses in groundwater from the up-gradient forestry land use. The most prevalent were nutrient losses during winter months when plant uptake is lower. However, outside of winter months, we observed a higher nitrate-nitrogen concentration (12 mg L$^{-1}$) after the trees were harvested, compared to 5.9 mg L$^{-1}$ when there was no harvesting, which we attribute to changing biogeochemical conditions. We used a novel hysteresis approach, comparing nitrate-nitrogen concentrations and groundwater levels after rainfall recharge to understand event scale variability. First flush events in winter had a larger area (more hysteresis) of 0.65 compared to an average area of 0.35 (less hysteresis) for subsequent events. Peak concentrations occurred earlier in events during 2021 (wetter) compared to 2020 (drier), signifying slower drainage pathways in years with less recharge. Through this analysis we also found evidence that the mobilisation of nitrate-nitrogen shifted from rainfall recharge to rising groundwater levels after the surface supply was depleted from successive recharge events. Finally, the nitrate-nitrogen load analysis indicates that leaching and export occurs in pulses, that discrete sampling cannot accurately characterise. For example, in 2021, over 80 percent of the exported load occurred during a quarter of the year and discharged when there were base flow conditions in the nearby Hurunui River. These findings have implications for forestry land management and the understanding of inorganic nitrogen dynamics in groundwater in response to rainfall recharge. Additionally, these insights may affect nitrate-nitrogen projections under climate change, where periods of drought and storm events are more frequent.

**Keywords**: Nitrate-nitrogen dynamics; UV nitrate sensor; forestry; hysteresis analysis; nutrient loads

# 1    Introduction

Inorganic nitrogen is a widespread contaminant in aquifers, surface and coastal waters (Paerl, 1997; Foster et al., 2013; Ward et al., 2018; Lall et al., 2020). The accumulation of excess inorganic nitrogen in both ground and surface waters presents risks to the ecological tolerance of aquatic species, and the provision of safe drinking water (Foster et al., 2013; Lall et al., 2020). Managing the effects of excess nutrient losses on water quality requires an understanding of the interaction between surface water and groundwater nutrient transfer at both catchment and local scales (Julian et al., 2016; Verhagen et al., 2022). However, it is difficult to accurately measure or quantify the movement of water between groundwater and surface water or vice versa, as the connection is diffuse (Sophocleous, 2002; Kalbus et al., 2006). The hydrological connection is further complicated by travel times along different flow paths in groundwater, as it leads to variable discharge rates (Ascott et al., 2017; McDowell et al., 2021). As such, reducing inorganic nitrogen-enriched recharge to rivers presents a challenge for local regulators that aspire to maintain or improve existing riverine water quality (Snelder et al., 2017; Snelder et al., 2018; Snelder et al., 2020).

Globally, nitrate-nitrogen concentrations in groundwater are increasing (Schlesinger, 2009), concomitant with expanding intensive agricultural land use (Abascal et al., 2022; Schulte-Uebbing et al., 2022; Daughney et al., 2023), which often dominates the discourse on inorganic nitrogen contamination (Shukla and Saxena, 2018; Bijay-Singh and Craswell, 2021). However, other types of land use change, such as plantation forestry, are also potential sources of nutrient losses (Kreutzweiser et al., 2008; Paré et al., 2016). The leaching of nutrients from exotic and native forestry is a source of inorganic nitrogen that has been widely studied throughout the world (Likens et al., 1970; Vitousek et al., 1979; Bechtold et al., 2003; Rothe and Mellert, 2004; Gundersen et al., 2006; Argerich et al., 2013) and in our New Zealand case study context (Dyck et al., 1983; Quinn and Stroud, 2001; Clinton et al., 2002; Parfitt et al., 2002; Davis, 2014; Baillie and Neary, 2015; Julian et al., 2016). Many studies on forestry inorganic nitrogen leaching report low nitrate-nitrogen concentration ranges in nearby streams and soil drainage water (Argerich et al., 2013; Davis, 2014) relative to agricultural land uses (Di and Cameron, 2002; Vuorenmaa et al., 2002), however, a range of factors can increase inorganic nitrogen leaching in forestry ecosystems. In forest ecosystems, the timing of inorganic nitrogen availability is determined through organic material undergoing phases of decomposition and immobilisation (Vitousek et al., 1982). When environmental conditions (e.g., rainfall, temperature and litter C:N ratio) are favourable for decomposition, mineralisation and nitrification processes can occur (Anaya et al., 2006; Likens et al., 1977). During periods when organic material is mineralised at a higher rate than plant uptake of inorganic nitrogen, the leaching of excess inorganic nitrogen can occur after rainfall (Mo et al., 2003).

Changes in nutrient uptake potential (Bechtold et al., 2003; Gundersen et al., 2006), hydrological conditions (Sebestyen et al., 2014), forest harvesting (Rosén and Lundmark-Thelin, 1987; Gundersen et al., 2006; Mupepele and Dormann, 2016) and subsequent changes in soil temperature (Dirnböck et al., 2016) and water balance (Bauhus and Bartsch, 1995) can all increase inorganic nitrogen availability. The seminal study in the Hubbard Brook catchment by Likens et al (1970) found that there was a pronounced change in stream nitrate-nitrogen concentrations from 0.2 to 18.5 mg L$^{-1}$, as a result of forestry harvesting and herbicide use, reducing nutrient uptake. These studies show that the degree of nutrient leaching depends on multiple biogeochemical factors (Green et al., 2023). Forestry practices where the whole tree is harvested or disturbed, and subsequent plant regrowth is suppressed have recorded the highest nutrient leaching rates (Likens et al., 1970; Gundersen et al., 2006; Mupepele and Dormann, 2016) Furthermore, studies investigating disturbed forestry systems have observed that nitrate-nitrogen concentrations in drainage water are typically higher than stream concentrations (Table 1) and have reported concentrations over 10 mg L$^{-1}$ (Vitousek and et al., 1979; Dyck et al., 1983).

**Table 1:** Nitrate-nitrogen (NO₃-N) concentrations from studies measuring baseline conditions or the effect of disturbance on forestry nutrient leaching. Where there was a change in the ecosystem during the study, the maximum nitrate-nitrogen value presented in the table occurred after disturbance.

| Location and study | Ecosystem change | NO₃-N range | Study Duration | Monitoring frequency |
|---|---|---|---|---|
| Denmark (Callesen et al., 1999) | Atmospheric deposition / Clear cutting | $0 - 51$ mg $L^{-1}$ drainage water | 1986-1993 | Annually and monthly |
| Michigan, USA (Iseman et al., 1999) | Clear cutting | $0 - 30$ mg $L^{-1}$ drainage water | 1991-1996 | Monthly |
| New Hampshire, USA (Likens et al., 1970) | Whole tree harvesting | $0.2 - 18.5$ mg $L^{-1}$ stream | 1963-1982 | Weekly |
| Wales (Neal et al., 2004) | Clear cutting | $0.5 - 3.4$ mg $L^{-1}$ stream and groundwater | 1983-2002 | Weekly |
| England (Reynolds et al., 1992) | Clear cutting | $1.3 - 4.3$ mg $L^{-1}$ stream | 1985-1991 | Fortnightly |
| Auckland, New Zealand (Smith et al., 1994) | Stem harvesting | $0.5 - 4$ mg $L^{-1}$ drainage water | 1986-1988 | Annually |
| Eyrewell, New Zealand (Davis et al., 2012) | No disturbance | 1.35 (mean) mg $L^{-1}$ drainage water | 2009-2011 | 6-weekly intervals |
| Bay of Plenty, New Zealand (Collier and Bowman, 2003) | No disturbance | $0.2 - 0.6$ mg $L^{-1}$ stream | 1996-1997 | Monthly |
| Bay of Plenty, New Zealand (Parfitt et al., 2002) | No disturbance | 3.5 mg $L^{-1}$ spring | 1996-2001 | Quarterly |
| Hawkes Bay, New Zealand (Fahey and Stansfield, 2006) | No disturbance | 0.15 mg $L^{-1}$ stream | 1995-2005 | Fortnightly |

NB: Some northern hemisphere studies consider that atmospheric deposition is an important factor for observed nitrate-nitrogen concentrations, in addition to forestry processes. Atmospheric deposition is not expected to be a significant contributor to nitrate-nitrogen concentrations in this study's context because of naturally low nitrate deposition in New Zealand (Meder et al., 1991; Nichol et al., 1997; Parfitt et al., 2006).

The temporal variation of nitrate-nitrogen concentrations within groundwater in forestry catchments is not as widely studied as intensive agriculture, and there are currently few studies that have used high-frequency UV nitrate sensors in the forestry setting (Pellerin et al., 2012). These high-resolution instruments provide an opportunity to investigate the variability of nutrient dynamics and leaching rates. With high-resolution data analysis, the temporal variability of nutrient losses to receiving environments can also be better constrained.

## 1.1 UV nitrate sensors and high-resolution data analysis techniques

UV nitrate sensors have been used to monitor nitrate-nitrogen concentrations in surface water, springs, wastewater and drinking water supplies (Pellerin et al., 2012; Miller et al., 2016; Burkitt et al., 2017; Miller et al., 2017; Burns et al., 2019; Hansen and Singh, 2018; Shi et al., 2022). More recently they have been applied to in-situ groundwater monitoring studies to elucidate temporal variability (Opsahl et al., 2017; Burbery et al., 2021; Haas et al., 2023; Husic et al., 2023). The use of UV nitrate sensors allows for reduced maintenance and calibration compared to methods that use reagents, while greatly increasing measurement frequency compared to discrete sampling programmes (Pellerin et al., 2013).

UV nitrate sensors measure the transmittance of light across a known path length (Pellerin et al., 2013; Huebsch et al., 2015). The amount of light transmitted is related logarithmically to the absorbance characteristics of the groundwater through the Beer-Lambert law (Huebsch et al., 2015). The Beer-Lambert law can be used to determine the concentration of nitrate-nitrogen in groundwater based on the amount of light at a specific wavelength transmitted from the light source to the detector and processor (Pellerin et al., 2013). However, water turbidity and coloured dissolved organic matter (CDOM) may also absorb or scatter light at similar wavelengths to nitrate-nitrogen and these concentrations also need to be accounted for as interference effects (Snazelle, 2016). Thus, UV nitrate sensors also measure the transmittance of light at wavelengths for common substances that cause interference and use algorithms to adjust a proportion of the absorbance rates to the nitrate-nitrogen concentration (Snazelle, 2016).

Using high-frequency UV nitrate sensors to monitor nitrate-nitrogen in groundwater can consequently provide new insights into sources, processes and pathways that are temporally variable. As the mechanisms that control inorganic nitrogen movement are also highly variable across catchments, high frequency monitoring can be applied to characterise a range of settings to gain new insights into inorganic nitrogen export through groundwater pathways (Liu et al., 2020; Burbery et al., 2021; Haas et al., 2023; Husic et al., 2023). Such temporally dense data provided by in situ UV nitrate sensors can then be used as the basis for time-series analysis techniques traditionally used in surface water studies that have not previously been applied in groundwater quality analysis.

Chemographs, for example, plot concentration (y-axis) over time (x-axis) and are a useful tool for understanding pulsing and dilution behaviour, whereas concentration (y-axis) over discharge (x-axis) plots reveal the timing and phasing of mobilisation from local or distal sources (Evans and Davies, 1998; White, 2002; Dupas et al., 2016; Liu et al., 2021). These concentration-discharge plots often present as either concordant phasing between concentration and water discharge (i.e., a linear, or non-linear trend), or as discordant phasing, whereby concentration varies with ascending or descending patterns in discharge. The latter may present as hysteretic curves that describe the relationship between concentration and discharge during discrete events (Evans and Davies, 1998).

Hysteresis can be described as source or transport limited behaviour. A clockwise loop describes where the source (or pathway length) of the solute is located nearby or is rapidly transported. Conversely, an anticlockwise loop indicates a distant source with a delay in solute transport to the monitoring site (Liu et al., 2021). In hydrological research, hysteresis analysis is traditionally applied to streams and rivers, where discharge measurements are more easily obtained (Lloyd et al., 2016; Vaughan et al., 2017; Baker and Showers, 2019). Where rainfall recharge is the driver of changes in groundwater levels and nitrate-nitrogen concentrations, hysteresis analysis can be used to identify the lag between these two variables following a recharge event. While several hysteresis studies have considered the effect of near stream groundwater on stream-derived hysteresis curves, high-frequency groundwater measurements have not been widely utilised (Aubert et al., 2013; Jacobs et al., 2018; Knapp et al., 2020; Winter et al., 2021; Gelmini et al., 2022).

Antecedent conditions play an important role in explaining the response of nitrate-nitrogen concentrations in groundwater after rainfall. The flushing of inorganic nitrogen after periods of drought can result in higher concentrations in groundwater from additional storage being mobilised in soil or the vadose zone (Van Metre et al., 2016; Opsahl et al., 2017; Leitner et al., 2020; Jutglar et al., 2021; Winter et al., 2023). Conversely, a more muted nitrate-nitrogen response may occur after consecutive recharge events temporarily exhaust the supply of nitrate-nitrogen (Wang et al., 2020; Yue et al., 2023). As a result, each recharge event has a distinctive nitrate-nitrogen response due to antecedent conditions. Hysteresis analysis in groundwater (i.e., examining concentration versus groundwater level) may help to describe the temporal variability of these recharge events.

The objectives of this study are to quantify event-based nitrate-nitrogen dynamics from a forestry-dominated land use as the principal source of inorganic nitrogen in groundwater. The study uses four years of high-frequency UV nitrate sensor data at a case study location in Canterbury, New Zealand. We investigate how high- frequency measurements of nitrate-nitrogen and groundwater levels along with the analysis techniques that these measurements enable, can help elucidate nitrate-nitrogen dynamics from source to discharge point. The specific analysis techniques that this study utilises includes hysteresis analysis of high-frequency nitrate-nitrogen concentrations and groundwater levels to understand how nitrate-nitrogen stores and pathways change during rainfall events, seasonally, and in response to broader variations in climate conditions. These data will also be used to refine estimates of nitrate-nitrogen loads and leaching by improving the understanding of temporal dynamics of groundwater nitrate-nitrogen discharges to riverine systems to inform freshwater management strategies.

## 2 Study Area

### 2.1 Geology and hydrogeology

The Culverden Basin of Canterbury, New Zealand is located in a transitional area between the continental collision (transform fault) of the Australian and Pacific plates (Alpine Fault) to the west and the Hikurangi subduction zone to the east (Armstrong, 2000; Barrell and Townsend, 2013; Brough, 2019). The rapid uplift from the compression of the Australian and Pacific plate collision (Early Miocene), resulted in large volumes of sediment eroding and subsequently infilling the developing basins (Armstrong, 2000). These Quaternary gravel deposits have created aquifers that are the main source of groundwater in the Culverden Basin and surrounding area (Poulsen, 2012).

Groundwater occurs in Quaternary age glacial outwash gravels (Burnham formation) from the Hurunui and Waiau Uwha Rivers (Poulsen, 2012) that are generally less than 50 metres thick (Armstrong, 2000). The alpine rivers in the Culverden Basin (Hurunui and Waiau Uwha) are incising the previously deposited alluvial fans as a result of continued uplift and lower rates of sediment deposition in post-glacial riverine environments (Armstrong, 2000). Due to the incising alpine rivers, groundwater in the surrounding alluvial terraces is often at a higher elevation than the height of these rivers. As a result, groundwater and land use are highly connected to the rivers that flow through the Culverden Basin, with groundwater vulnerable to nitrate-nitrogen accumulation in areas of limited recharge (Abraham and Hanson, 2006; Knottenbelt, 2023).

Mean annual precipitation in the Culverden Basin is 576 mm per year (1981-2010), with nearly half (46 percent) of rainfall over that period falling in winter months (Macara, 2016). On average the Culverden Basin experiences 99 days of rainfall per year (more than 0.1 mm) and 82 days where more than 1 mm of rainfall fell (Macara, 2016). During summer months, potential evapotranspiration can be high, with an annual average (Penman PET) of 845 mm from 2018 to 2024 (NIWA. 2024). This indicates there can be a soil moisture deficit during summer months.

### 2.2 Monitoring site description

A 24 metre deep well was installed downgradient and adjacent to a 25 km$^2$ area of *Pinus radiata* forest in the Culverden Basin, approximately 1.5 kilometres from the true left bank of the Hurunui River (172.792, -42.844, WGS 84). The drill core showed the local stratigraphy is comprised of a very thin layer of silty pallic soil (~0.30 m) with distinct alluvial deposits visible at the surface. Pallic soils in this region are known for their low water holding capacity, tendency to crack during summer and low carbon content relative to other soil types in New

Zealand (Hewitt et al., 2021). As such, these soils are recognised as being physiographically poor attenuators of inorganic nitrogen and tend to promote the leaching of excess nutrients (Hewitt et al., 2021). The thin stony soil and highly permeable gravel aquifer at the site results in rapid changes to groundwater conditions after rainfall. Immediately upgradient is an area of active pine (*Pinus radiata*) forestry (Fig. 1). During the study period (2020 to the end of 2023), sections of pine forest were logged and cleared. Through satellite imagery, it was observed

that in areas where logging has occurred there was no significant plant regrowth. Downgradient of the site is an area that has been converted from forestry to irrigated beef. The land use of the broader region is dominated by pastoral agriculture, particularly beef and dairy grazing.

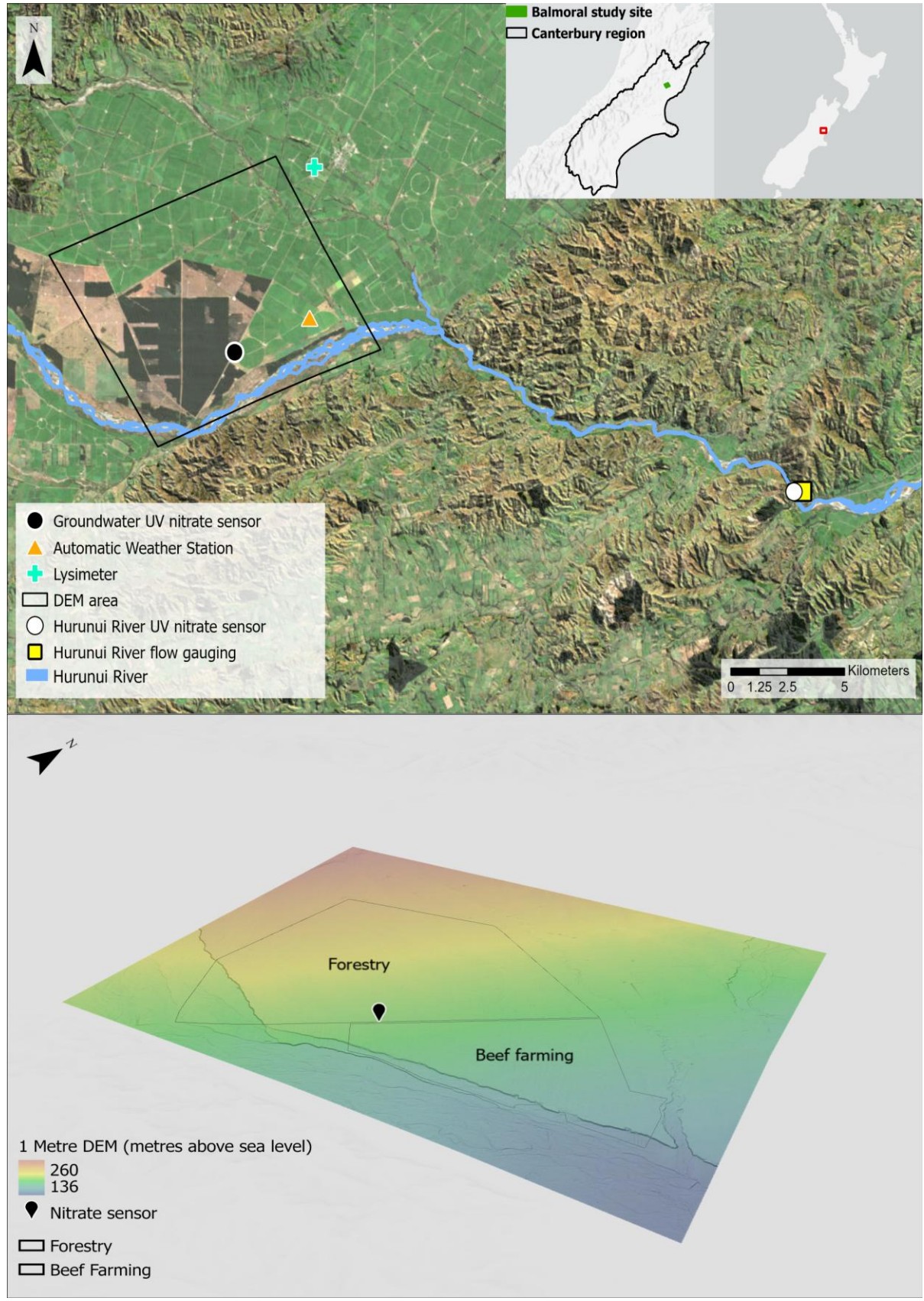

**Figure 1**: Study area at Balmoral in the Culverden Basin and the locations of groundwater and Hurunui River monitoring sites (ESRI, 2024). The digital elevation model indicates that the direction of groundwater flow is from the northwest to southeast.

### 3 Methods

#### 3.1 Monitoring site setup

A TriOS NICO UV nitrate sensor was installed to monitor nitrate-nitrogen concentrations in groundwater at a depth of 20 metres, which is below the lowest groundwater level. Results were reported as nitrate-nitrogen in milligrams per litre. The path length was set at 5 mm to account for the wide range of nitrate-nitrogen concentrations observed at the site. However, the path length and associated accuracy (approximately $\pm$ 5% + 0.5 mg L$^{-1}$) of the UV nitrate sensor cannot be optimised for all observed nitrate-nitrogen concentrations and as a

result, measurements of nitrate-nitrogen can be overestimated. The high-frequency measurements were recorded at 15-minute intervals with a xenon light source of 212 nm wavelength. Interference absorption was also measured at 254 nm for organic compounds and at 360 nm for turbidity. The TriOS NICO also derives and outputs the Sensor Quality Index (SQI), a rating of the measurement quality used for quality assurance. Due to the diameter of the well, the TriOS NICO UV nitrate sensor was vertically suspended. A Zebratech wiper was fitted to the

TriOS NICO to reduce interference from turbidity settling on the lens or biofouling.

Groundwater levels were measured at 15-minute intervals through a float and weight system (Unidata 6541). Rainfall volume was also measured at 15-minute intervals by a tipping bucket rain gauge (Davis AeroCone Rain Collector). Additionally, a 12 V submersible pump (Proactive Super Twister) was permanently installed in the well to improve access to discrete groundwater quality sampling.

Nitrate-nitrogen standards were periodically (3 monthly) used to check the precision of the TriOS NICO at different concentrations. The accuracy of the UV nitrate sensor measurements was also compared to laboratory tested validation measurements of discrete nitrate-nitrogen samples (Fig. 2). Validation samples also included parameters that could cause interference, such as dissolved organic carbon and turbidity. Regular quarterly groundwater monitoring also occurs at the site, which involves a full suite of parameters including metals,

nutrients and bacteria.

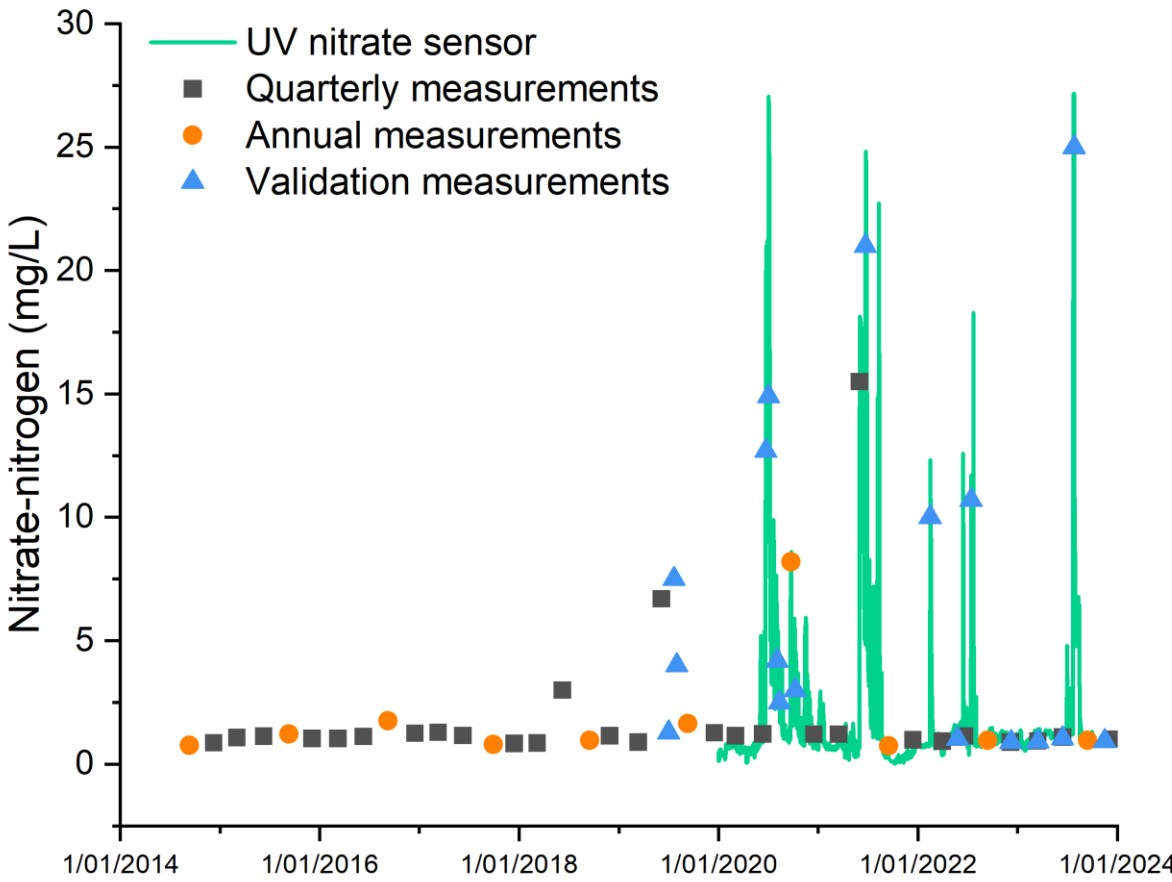

**Figure 2:** Groundwater nitrate-nitrogen concentrations at the monitoring site from the high-frequency UV nitrate sensor and discrete annual, quarterly and validation samples. Pre-2020 validation samples are from when the UV nitrate sensor was being tested.

The UV nitrate sensor measurements were compared to discrete validation samples to check the accuracy of the UV nitrate sensor (Fig. 3). The coefficient of determination ($R^2$) value of 0.93 indicated a strong regression line of best fit and that both the discrete and high-frequency measurements are similar ($n = 34$). The best fit between UV nitrate sensor measurements and validation samples occurs when nitrate-nitrogen concentrations are less than 5 mg L$^{-1}$. Figure 3 indicates that the UV nitrate sensor can overestimate (y = 1.13x ± 0.058) high concentrations of nitrate-nitrogen (i.e., the difference between discrete and high-frequency data).

The post processing of data from the UV nitrate sensor involved outlier and step change detection as well as gap-filling when data was not recorded. Outliers were identified by comparing high frequency and discrete samples. We adjusted the UV nitrate sensor data where the difference between high frequency data and discrete samples was greater than the accuracy of the UV nitrate sensor. Step changes due to interference effects and no measurements due to loss of signal or an error were also identified. Linear interpolation was used to fill gaps in data from outliers, step changes and no measurements.

.

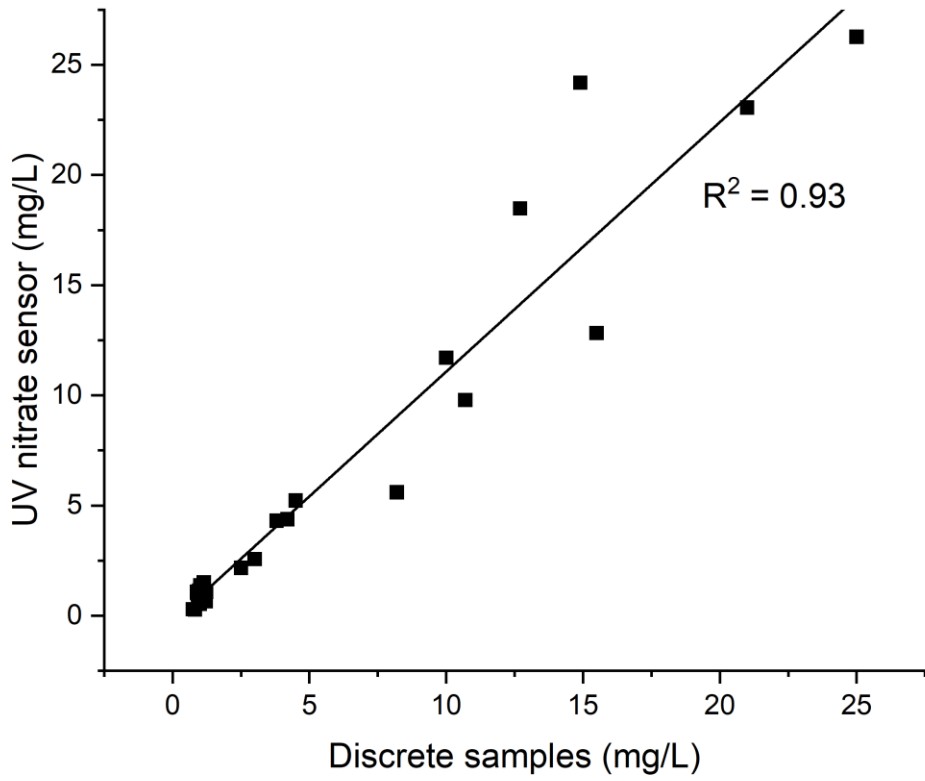

**Figure 3:** The relationship between nitrate-nitrogen measurements from the UV nitrate sensor and concurrent discrete samples.

### 3.2 Rainfall recharge

A 1-D land surface recharge model was calibrated using nearby lysimeter data to determine when rainfall recharge occurred. The parameters used in the land surface recharge model included precipitation, and Penman PET data sourced from a nearby weather station (Fig. 1). Soil water capacity (86.0 mm), an evaporation reduction function (10.0), and a drainage threshold (50.0) were derived from the nearby lysimeter (Fig. 1) (Bidwell and Burbery, 2011), while the crop factor (0.77) was sourced from Allen et al (1998). In this model, rainfall recharge occurs when rainfall is above the soil water storage capacity. Evaporation, plant uptake and runoff are factors that reduce soil water storage. River recharge was not considered to be a strong control on groundwater levels in the model because of the site's mean groundwater level (173 m asl) compared to the elevation of the Hurunui River (162 m asl). The abstraction of groundwater was also not considered when interpreting groundwater levels as there are no nearby wells that extract groundwater.

### 3.3 Hurunui River flow and Hurunui River nitrate-nitrogen

We investigated nitrate-nitrogen concentrations and flow data in the Hurunui River to understand the impact of the nitrate-nitrogen load from groundwater. The flow of the Hurunui River was measured at the State Highway One bridge site (Fig. 1) using a rating curve to relate gauged flow and water level stage height. The stage height was measured at 5-minute intervals, from which the mean was calculated and reported as a daily flow average. The high frequency nitrate-nitrogen concentration in the Hurunui River were measured using a TriOS NICO, at the State Highway One bridge, approximately 30 kilometres downstream of the groundwater monitoring site (Fig. 1). The depth of the UV nitrate sensor in the Hurunui River was adjusted to account for summer and winter seasons. The UV nitrate sensor measured concentrations of nitrate-nitrogen at 15-minute intervals, which were calibrated by monthly validation samples. Due to this UV nitrate sensor being in a braided river, the sensor

experienced additional interference effects and required more maintenance. There were periods where no data was available because the sensor was out of the water for calibration, repairs or the data was discarded because the SQI and discrete samples indicated the data was not accurate.

### 3.4 Regional groundwater levels as an indicator of climate variation

Regional groundwater levels across Canterbury were analysed as an indicator of climate variation (rainfall recharge) over the study period. The purpose of this analysis is to show the climate variation that controlled rainfall recharge over the study period. Monitoring wells with at least 10 years of groundwater levels in the Canterbury region were used to show monthly groundwater level percentiles over time. Groundwater levels were categorised as very low (bottom 10%), low (10-25%), average (25% to 75%), high (75-90%) or very high (top 10%) and then aggregated to determine the number of Canterbury wells in these percentiles for each month.

### 3.5 Hysteresis analysis

Increases in nitrate-nitrogen and the associated groundwater level were graphed for each event between 2020 and the end of 2023. To determine when hysteresis analysis should occur, we considered the accuracy of the UV nitrate sensor and groundwater levels as well as the variation in baseline conditions. For these reasons we only analysed events with an increase in nitrate-nitrogen over 1 mg L$^{-1}$ (25 events). To visually show the rate of change in nitrate-nitrogen and groundwater levels, the hysteresis curves were split into five equal sections, each represented by a different colour on the graph. The hysteresis curves were categorised as having hysteresis (or not) and the direction of hysteresis. As multiple recharge events often occurred consecutively, the nitrate-nitrogen concentration or groundwater level did not always recover to pre-event conditions. Therefore, the next event began at the lowest nitrate-nitrogen concentration after the previous event.

We used the Hysteresis, Area, Residual and Peak (HARP) analysis method in R (Roberts et al., 2023) to compare metrics on hysteresis curve area, residual (difference in concentration between the rising and falling limbs) and the proportion of time into the event that peak groundwater level and nitrate-nitrogen concentration were observed in 2020 (15 events) and 2021 (10 events). The analysis in R included truncating data to form a closed loop and the normalisation of groundwater levels and nitrate-nitrogen concentrations. The hysteresis area was represented as a decimal value between 0 and 1, with larger values indicating more hysteresis. The time to peak nitrate-nitrogen or groundwater level are percentages that indicate the time along the hysteresis curve (or event) where the maximum values occur (Roberts et al., 2023). The residual analysis was used to determine if the environment shifted to new baseline concentrations after recharge events. The results of the residual analysis were not analysed further due to nitrate-nitrogen concentrations returning to pre-event conditions. Recharge events were not analysed using HARP in 2022 (6 events) and 2023 (2 events) as they did not present as hysteresis curves. The R script used to process the hysteresis data is available on Github (Roberts, 2022).

The hysteresis type analysis identifies the storage, mobilisation and flow pathways of nitrate-nitrogen in the catchment that contribute to the observed concentrations at the monitoring well (Evans and Davies, 1998; Knapp et al., 2020 Liu et al., 2021). Clockwise hysteresis curves indicate that the source of nitrate-nitrogen is nearby (short pathway), rapidly mobilised and is progressively depleted along the hysteresis curve (Husic et al., 2023). In contrast, anticlockwise hysteresis curves represent longer flow pathways, less immediately mobile stores of nitrate-nitrogen, where the groundwater level change is initially greater (Roberts et al., 2023). Complex or 'figure of eight' hysteresis curves indicate that the source or flow path of nitrate-nitrogen changes during the event (Baker and Showers, 2019; Liu et al., 2021). The area of the hysteresis curve is a metric that can be used to compare events. A larger area shows a greater difference in the rate of change between nitrate-nitrogen and groundwater levels (Roberts et al., 2023).

### 3.6  Catchment nitrate-nitrogen load analysis

The annual nitrate-nitrogen load from groundwater to the Hurunui River between 2020 and 2023 from the site was calculated using trapezoidal integration of the high-frequency UV nitrate sensor data (Richards, 1998). The timing of the nitrate-nitrogen load was analysed and compared to flow as well as nitrate-nitrogen concentrations in the Hurunui River. The groundwater discharge component was calculated using Darcy's Law in Eq. (1), as described below. We modified the hydraulic conductivity units to be metres per 15 minutes to align with the
recorded nitrate-nitrogen data interval. The saturated thickness of the aquifer was calculated using the high-frequency depth to groundwater (highest elevation) and the elevation of the Hurunui River (base). An upward gradient was also observed in nearby deeper wells, further supporting the sub-horizontal flow of nutrients to the Hurunui River. The hydraulic gradient was estimated using data from previous studies (Thomas and Veendrick, 2018) in the area. The hydraulic gradient was varied in the timeseries for increasing groundwater levels (highest),
decreasing groundwater levels and stable groundwater levels (lowest).

$$Q = KiA \tag{1}$$

*Where*:

Q = Groundwater flow ($m^3$ 15 minutes$^{-1}$)

K = hydraulic conductivity (m 15 minutes$^{-1}$)

i = hydraulic gradient

A = cross sectional area of groundwater flow ($m^2$)

To calculate the nitrate load from the area upgradient of the monitoring borehole, the scipy.integrate python sub package was used to calculate the area under the curve of the high-frequency UV nitrate sensor data. Trapezoidal integration was used to approximate the area because of the UV nitrate sensor data characteristics, including a
high number of linearly spaced intervals, lack of symmetry and data smoothness. The area under the curve for the high-frequency nitrate-nitrogen data was calculated to give 15-minute intervals of nitrate-nitrogen in milligrams per litre (mg L$^{-1}$ 15 minutes$^{-1}$).

To calculate the volume of water moving in a horizontal direction from the aquifer to the Hurunui River, a time variable Darcy's Law equation was used at 15-minute intervals. The hydraulic gradient and saturated thickness
were varied based on the high-frequency groundwater levels. Due to the upward gradient in the adjacent deep well (screened at 98 to 101 m bgl), we assumed that a high proportion of the shallow groundwater moves horizontally, discharging to the Hurunui River. Therefore, the application of Darcy's Law was deemed to be appropriate for this scenario. As the hydraulic conductivity of the aquifer is heterogeneous and the gradient also varies, we used a range of potential hydraulic conductivity (Freeze and Cherry, 1979) and hydraulic gradient
values to understand variation in the load calculation. This resulted in a 25 percent increase or decrease in exported nitrate-nitrogen (Table 2).

The calculated volume of groundwater discharged from the aquifer at 15-minute intervals was multiplied by the concentration of nitrate-nitrogen per 15-minute interval and summed to give the total annual load of nitrate-nitrogen, as shown in Eq. (2). Statistics on the timing of the nitrate-nitrogen flux were calculated by integrating
specific time periods and comparing values to the total exported load.

$$Load = K \sum_{i=1}^{n} c_i q_i \frac{1}{2}(t_{i+1} - t_{i-1}) \qquad (2)$$

*Where*:

K = unit conversion constant

i = sample in the *i*-th position

$c_i$ = concentration (mg L$^{-1}$ 15-minutes$^{-1}$)

$q_i$ = discharge (m$^3$ 15-minutes$^{-1}$)

t = time period for the *i*-th sample

## 4    Results

### 4.1    Factors that influenced high rates of nitrate-nitrogen leaching

The observed pulses of nitrate-nitrogen during the study period reflect complex interactions between forestry harvesting and hydrological conditions. To understand the drivers of nitrate leaching and storage, we examined the timing of forestry harvesting and patterns of rainfall recharge. There was a strong relationship between prior forestry harvesting and rainfall recharge driving nitrate-nitrogen mobilisation. In contrast, low nitrate-nitrogen concentrations were consistently observed during dry periods, aligning with phases of nitrate-nitrogen storage and low groundwater levels. These findings suggest that both land use changes and hydrological variability play a critical role in understanding pulses of nitrate-nitrogen, which we explore in detail below.

### 4.1.1 Land cover change

Forestry cover reduced during the study period (Fig. 4). While most of the forestry harvesting was gradual, we identified significant changes in forestry cover through aerial imagery observations and time series analysis. There were three major reductions in forestry cover. One occurred before the UV nitrate sensor was installed in 2018. The other two events occurred in October 2021 and January 2022. All three forestry harvesting events were located close to the monitoring site.

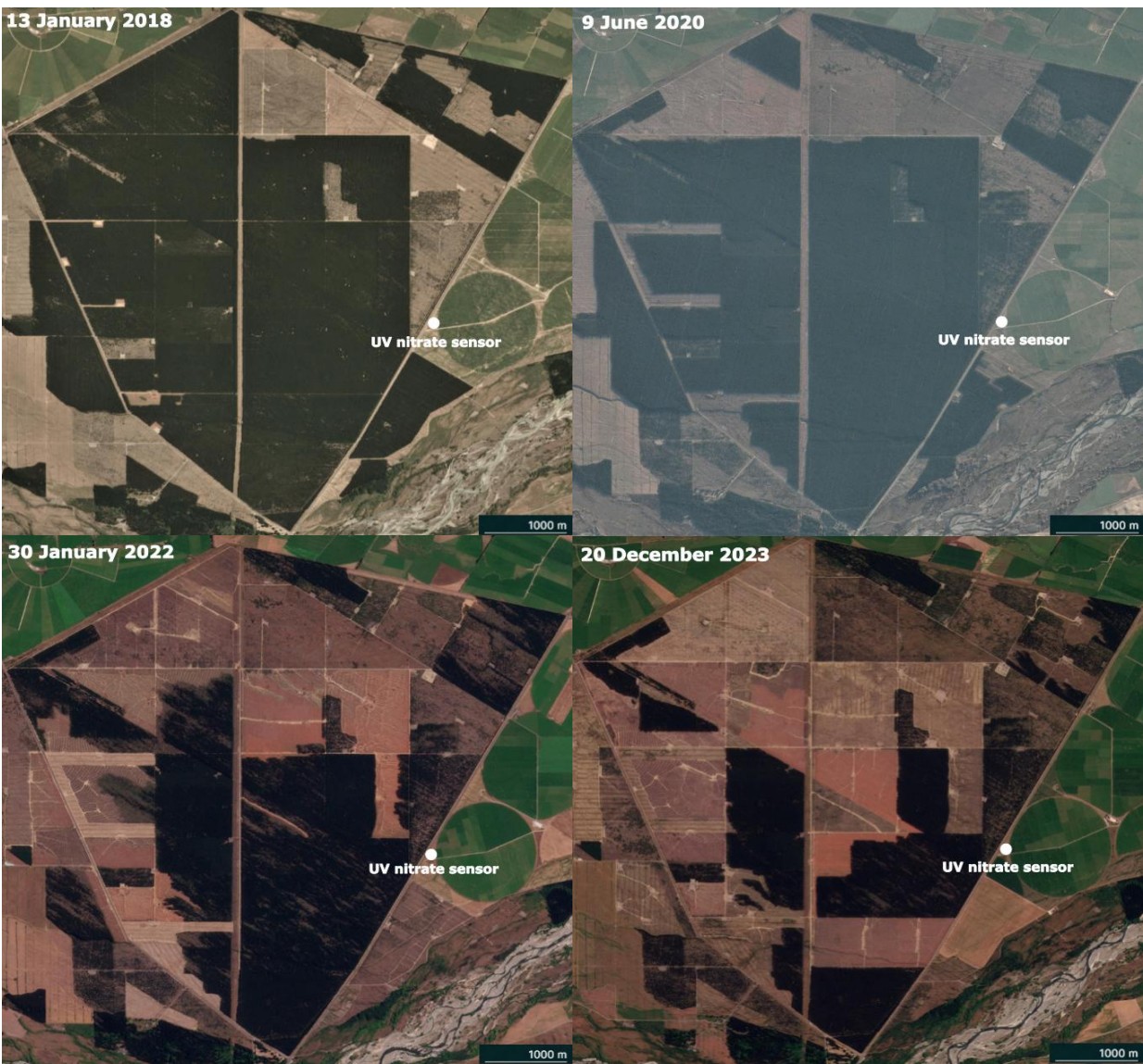

**Figure 4:** Land use cover changes from daily PlanetScope imagery (3 m resolution) showing the reduction in forestry cover from 2018 to 2023 (Planet, 2024)

### 4.1.2 Periods of low recharge

Groundwater levels, as a record of climate conditions prior to and during the study period, indicate that there were two distinct climate types (Fig. 5). From the start of 2020 to June 2021 there was low rainfall recharge in Canterbury (Fig. 5). These hydrological conditions were also observed at the monitoring site at Balmoral. In May 2021, over 80 percent of monitoring wells (including Balmoral) had groundwater levels that were 'low' or 'very low' compared to other measurements in the same month. After June 2021, groundwater levels had recovered by 2022, with less than 20 percent of wells recording 'low' or 'very low' groundwater levels.

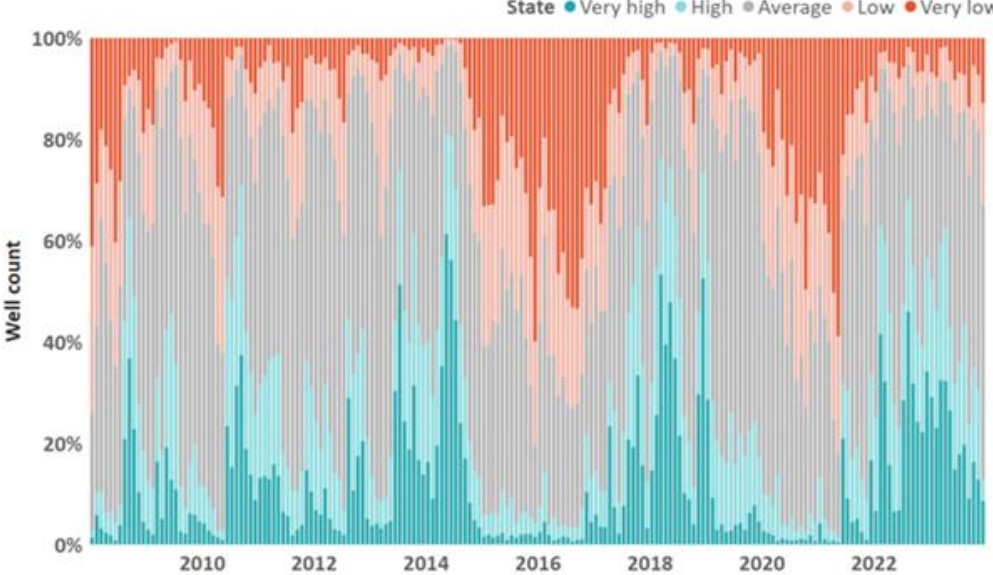

**Figure 5:** Canterbury groundwater levels as an indicator of recharge conditions at Balmoral (Fig. 1). An increase in low or very low groundwater levels (red) indicate reduced precipitation and low recharge. An increase in high or very high groundwater levels (blue) indicate periods of increased precipitation.

### 4.2 Nitrate-nitrogen dynamics

The highest concentrations of nitrate-nitrogen in groundwater were observed during winter months (June – August), which coincided with the majority of rainfall recharge. Nitrate-nitrogen concentrations showed a strong flushing response to rainfall recharge (Fig. 6) that resulted in elevated concentrations for short periods of time. The average duration of elevated nitrate-nitrogen concentrations was 8 days (n=33), before returning to baseline conditions. When rainfall recharge or increases in groundwater levels did not occur, nitrate-nitrogen concentrations in groundwater were below 1 mg L$^{-1}$. Apart from winter months, these conditions were present for most of the study period.

Due to low prior rainfall, groundwater levels were the lowest during the study period in 2020 and continued to be low until June 2021, when a large storm event (100 mm of rainfall over two days) resulted in significant rainfall recharge (Fig. 6c). After June 2021, groundwater levels continued to increase in response to successive rainfall recharge events. Outside of winter months, rainfall recharge also occurred in November 2020, and February 2022. For November 2020, the highest nitrate-nitrogen concentration was 5.9 mg L$^{-1}$ because of rainfall recharge. In February 2022, however, the highest observed nitrate-nitrogen concentration was 12 mg L$^{-1}$, which coincided with prior forestry harvesting and intense rainfall.

Nitrate-nitrogen concentrations in the Hurunui River showed concordant increases with flow and rainfall. Delayed peaks of nitrate-nitrogen concentrations in the Hurunui River were also observed after rainfall events. These delayed peaks of elevated nitrate-nitrogen concentrations occurred during base flow conditions. Data was missing in the Hurunui River nitrate-nitrogen timeseries due to maintenance and interference effects.

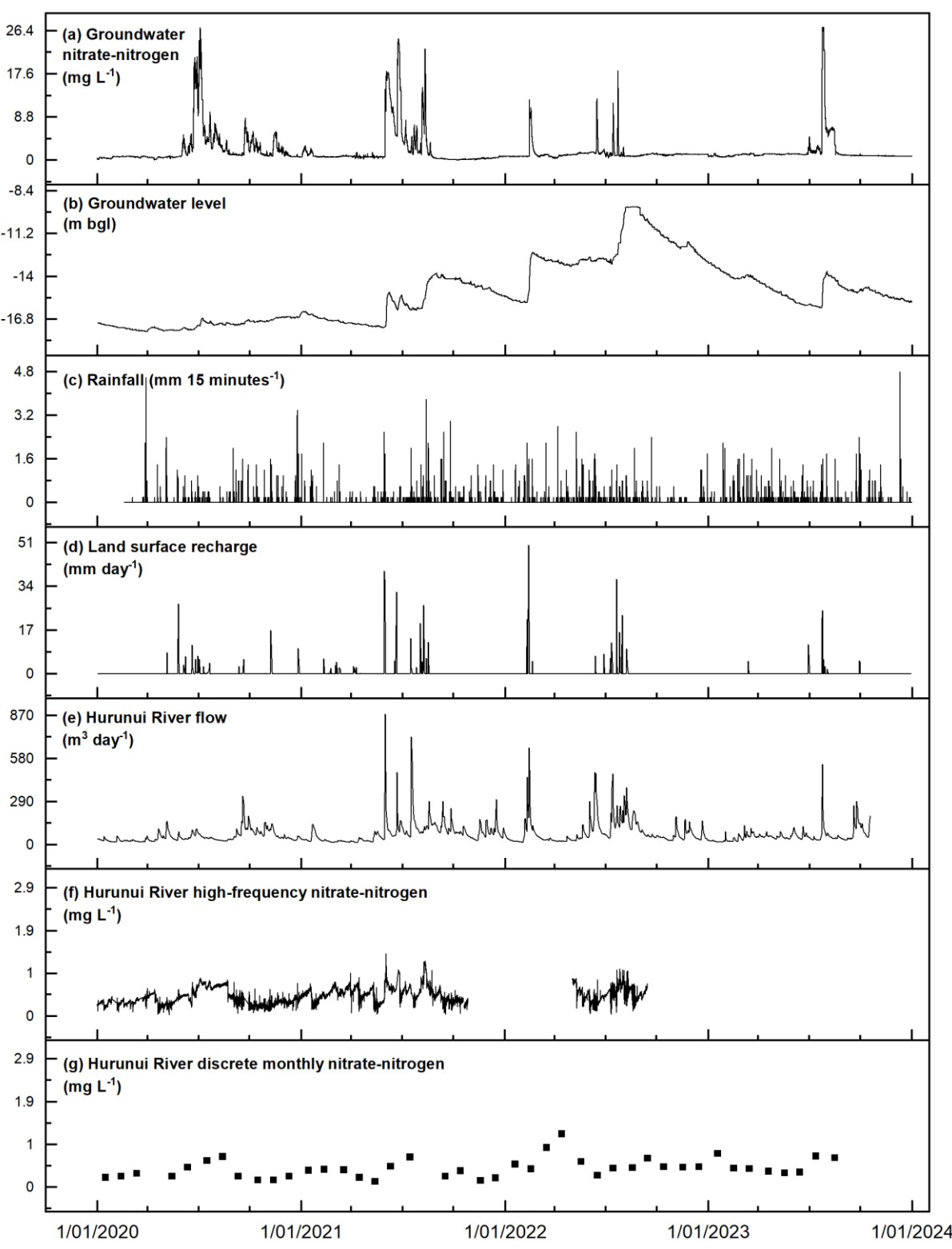

**Figure 6:** Timeseries of nitrate-nitrogen concentrations in groundwater at the site and hydrological drivers (rainfall and land surface recharge). The Hurunui River data shows fluctuations in flow and nitrate-nitrogen concentrations.

### 4.3 Hysteresis analysis using high-frequency nitrate-nitrogen and groundwater level measurements

We found that the relationship between nitrate-nitrogen and groundwater levels varied between years based on the number and magnitude of prior recharge events and interannual groundwater level trends (Fig. 7 and Fig. 8). Large rainfall recharge events resulted in clockwise (source limited) hysteresis curves and higher concentrations of observed nitrate-nitrogen. In 2020, there were periods of receding groundwater levels that resulted in increasing nitrate-nitrogen with declining groundwater levels after recharge (Fig. 7). For the last event at the end of winter in 2021, there was a positive linear relationship between increases and decreases in groundwater level and nitrate-nitrogen (Fig. 8). In 2022 and 2023 all recharge events produced clockwise loops that were not complete due to an increasing trend in groundwater levels. All graphs showing the relationship between groundwater levels and nitrate-nitrogen concentrations are available in the supplementary information.

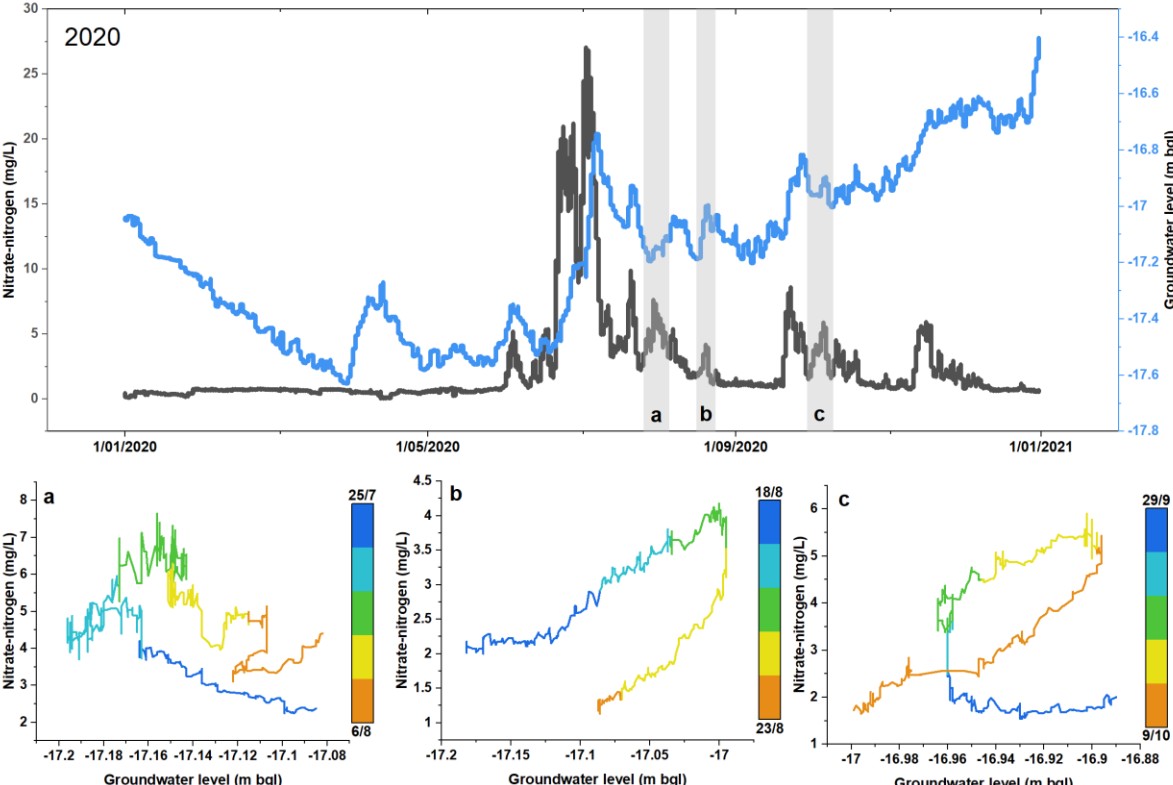

**Figure 7:** The selected hysteresis curves from 2020 show the increased variability in hysteresis curves observed during smaller recharge events. Graph a shows an increase in nitrate-nitrogen while there were receding groundwater levels. Graphs b and c are more typical clockwise hysteresis curves. All 15 hysteresis curves from 2020 are available in the supplementary information.

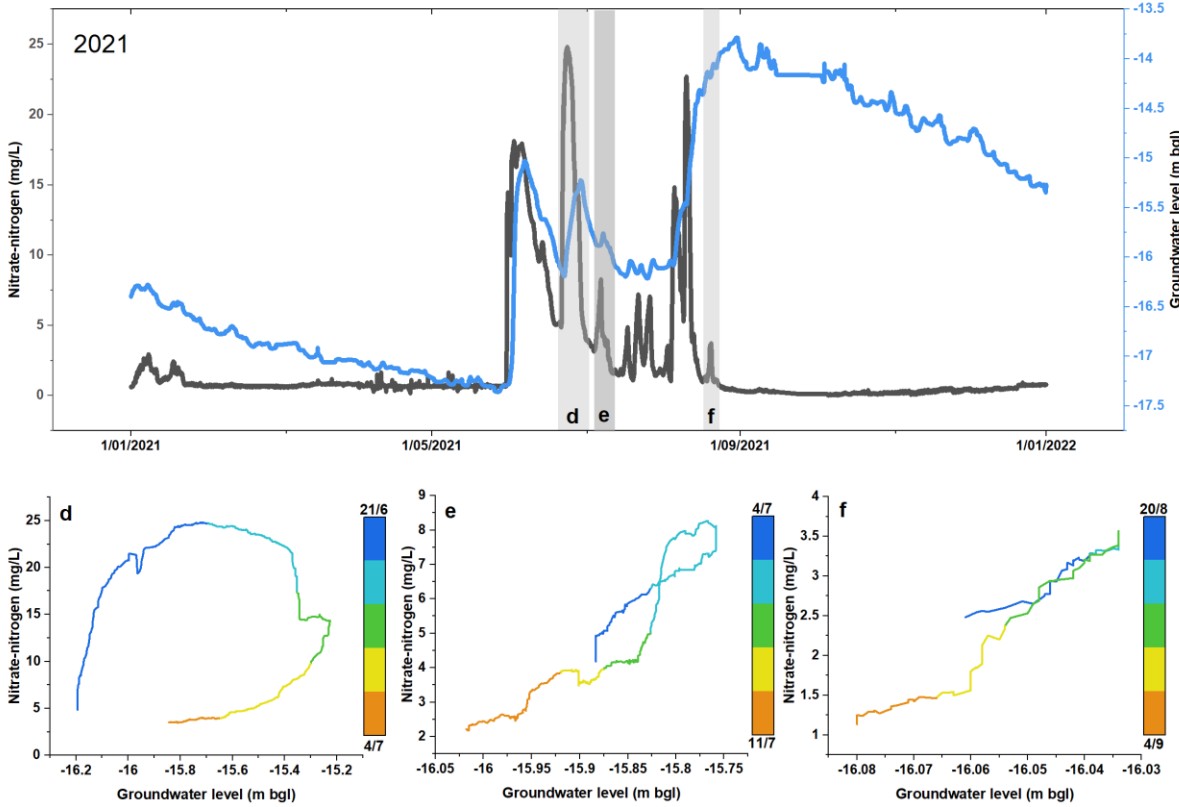

**Figure 8:** First flush events in 2021 had a large degree of hysteresis, shown by the area of the hysteresis curve (d). Subsequent events showed lesser degrees of hysteresis (e), with the last event showing a linear relationship between groundwater level and nitrate-nitrogen concentration after a significant rise in groundwater levels (f). All 10 hysteresis curves from 2021 are available in the supplementary information.

HARP analysis indicated the average hysteresis curve area of first flush events was larger at 0.65 compared to an average of 0.35 for subsequent events. The average proportion of time to reach peak groundwater level and nitrate-nitrogen concentration during events was lower in 2021 at 50% and 55% of events, compared to 2020, at 77% and 70% (Fig. 9). We compared the correlation (Pearson's *r*) of rainfall volume 10 days prior to an event and time to peak nitrate-nitrogen concentrations and groundwater levels between 2020 and 2021. In 2021, there was a moderately strong negative relationship between 10-day prior rainfall volume and peak nitrate-nitrogen (-0.65) and peak groundwater level (-0.61), indicating peaks occurred earlier in the event for larger rainfall events. There was no significant relationship between nitrate-nitrogen concentrations and prior rainfall volumes in 2020.

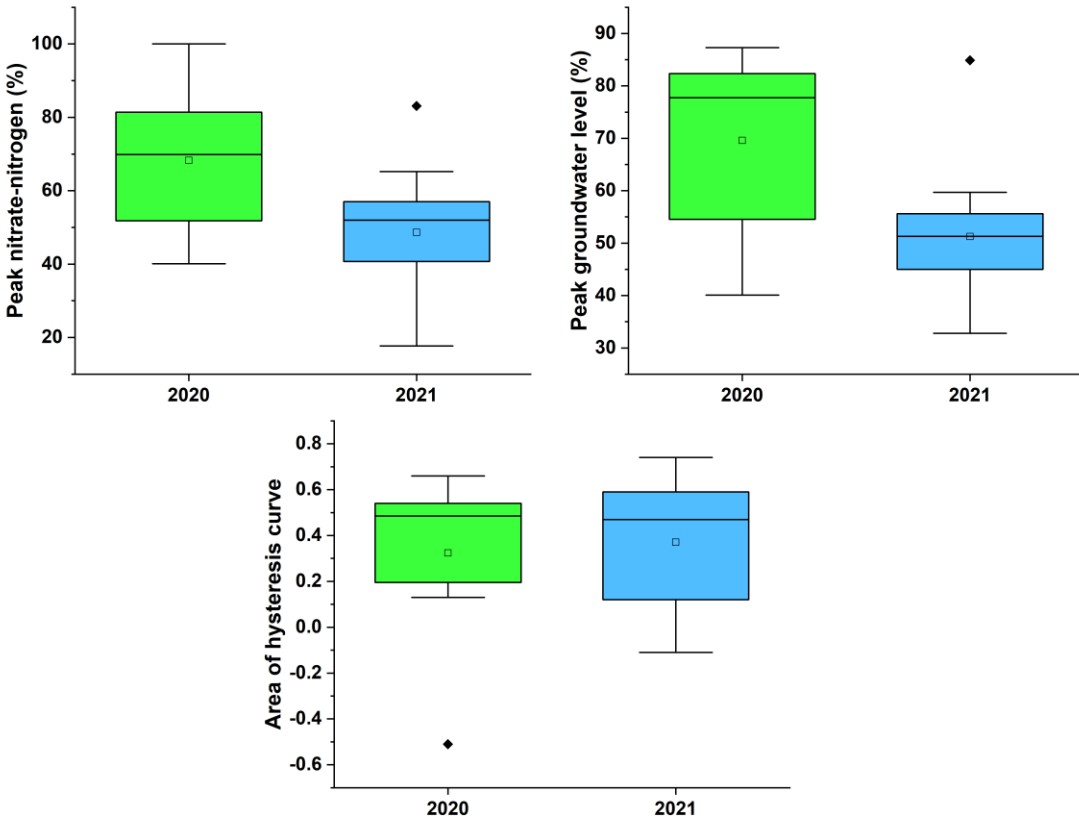

**Figure 9:** Boxplots from the HARP analysis of the percentage of time taken during recharge events to reach peak nitrate-nitrogen concentration, peak groundwater level and the area of the hysteresis curve during 2020 (15 events) and 2021 (10 events).

## 4.4 Nitrate-nitrogen load from groundwater to the Hurunui River

Pulses of high nitrate-nitrogen concentrations were observed in groundwater after rainfall recharge in winter months. The highest concentrations of nitrate-nitrogen (up to 26 mg $L^{-1}$) in groundwater occurred in winter months (June to August), with the winter median concentration over four years generally between 2.4 mg $L^{-1}$ and 5.7 mg $L^{-1}$ (Fig. 6a). In late June the daily median concentrations were between 11.3 and 18 mg $L^{-1}$. During summer, autumn and spring months, daily median nitrate-nitrogen concentrations were generally below 2.4 mg $L^{-1}$.

The nitrate-nitrogen load from groundwater was found to be highest during 2020, with an estimated 7.25 t $yr^{-1}$ exported to the Hurunui River (Table 2). There were higher intensity rainfall recharge events in 2021 and more rainfall recharge (265 mm) but the nitrate-nitrogen exported was slightly less than 2020 at 6.91 t $yr^{-1}$. There was less nitrate-nitrogen exported from groundwater to the Hurunui River in 2022 and 2023 at 5.97 t $yr^{-1}$ and 6.88 t $yr^{-1}$, respectively. We also calculated the exported load using the annual and quarterly measurements, which over- (or under-) estimated the load depending on whether they measured the winter pulses of nitrate-nitrogen (Figure 2) (Table 3).

**Table 2:** Estimated nitrate-nitrogen load per year from the UV nitrate sensor and rainfall recharge parameters from 2020 to 2023.

| Year | Nitrate-nitrogen export (t $yr^{-1}$) | Nitrate-nitrogen export per hectare (kg $ha^{-1}$) (1500 ha) | Rainfall (mm $yr^{-1}$) | Rainfall recharge (mm $yr^{-1}$) |
|---|---|---|---|---|
| 2020 | 7.25±1.81 | 4.83 | 601 | 150 |
| 2021 | 6.91±1.72 | 4.61 | 592 | 265 |
| 2022 | 5.97±1.49 | 3.98 | 762 | 249 |
| 2023 | 6.88±1.72 | 4.58 | 593 | 89 |

**Table 3:** Comparison of estimated nitrate-nitrogen loads from high-frequency, annual and quarterly discrete measurements.

| Year | Nitrate-nitrogen export (t $yr^{-1}$) (UV nitrate sensor) | Annual measurements (t $yr^{-1}$) | Quarterly measurements (t $yr^{-1}$) |
|---|---|---|---|
| 2020 | 7.25 | 23.4 | 8.73 |
| 2021 | 6.91 | 1.97 | 18.1 |
| 2022 | 5.97 | 3.90 | 4.27 |
| 2023 | 6.88 | 3.53 | 3.60 |

The majority of nitrate-nitrogen leaching occurred immediately after rainfall recharge for short time periods. In 2020, 50 percent of the total nitrate-nitrogen that was leached occurred during 13 percent of the year as a result of the observed nitrate-nitrogen spikes after rainfall. In 2021, the rapid leaching of nitrate-nitrogen was also observed as 80 percent of the total nitrate-nitrogen was exported during the heavy rainfall events in winter months, which accounted for 25 percent of 2021. Short pulses of leaching were also observed in 2023. Due to the short duration of high nitrate-nitrogen concentrations in 2022, only 10 percent of the total nitrate-nitrogen was leached during each observed pulse. As a result, there was more nitrate-nitrogen exported outside of the observed pulses in 2022.

## 5    Discussion

### 5.1    Drivers of nitrate-nitrogen dynamics

The presented results show that nitrate-nitrogen concentrations in groundwater varied in accordance with seasonal, climatic and disturbance-based changes in nutrient availability. We observed large pulses of nitrate-nitrogen in groundwater during every winter period (June to August) of the study, with a maximum concentration of 26 mg L$^{-1}$. In winter biological activity decreases, reducing nutrient uptake, which in turn increases the labile pool of excess nitrate-nitrogen (Gundersen et al., 2006; Davis, 2014). We infer that the reduced nutrient demand, increased rainfall and saturated soil conditions induced higher nutrient losses from the forest soils during winter (Bechtold et al., 2003; Sebestyen et al., 2014).

The decrease in forestry cover over the study period was a driver of high nitrate-nitrogen concentrations (Rosén and Lundmark-Thelin, 1987; Sebestyen et al., 2014). Notable harvesting occurred in October 2021 and in January 2022 (Fig. 4). After forestry harvesting, we observed a larger pulse of nitrate-nitrogen outside of winter months. In February 2022 (directly after the forestry harvesting), the maximum concentration was 12 mg L$^{-1}$ compared to 5.9 mg L$^{-1}$ in November 2020 when prior forestry harvesting did not occur. The larger volume of rainfall (75 mm) during February 2022 compared to November 2020 (35 mm), was also a factor in the higher observed nitrate-nitrogen concentration. This indicates there was more nitrate-nitrogen available after harvesting practices (Bauhus and Bartsch, 1995). There was also more nitrate-nitrogen available during first flush events (higher nitrate-nitrogen concentrations) at the start of winter, which were exacerbated in 2021 by the low recharge conditions observed in 2020 and early 2021 (Fig. 5). Nitrate-nitrogen leaching continued after harvesting because of windrowed debris and the limited regrowth of plants, post harvesting that could uptake nutrients (Likens et al., 1970; Slesak et al., 2009; Devine et al., 2012).

We identified that the relative importance of biogeochemical processes, such as the mineralisation and immobilisation of nitrogen shifted to the hydrological mobilisation of nitrogen after the intense rainfall event in May 2021. We infer that the catchment's ability to uptake nitrogen was reduced during this dry period (Winter et al., 2023) and was then continually disrupted due to forestry harvesting and the suppressed regrowth of understorey plants (Green et al., 2023). These conditions led to higher-than-expected nitrate-nitrogen concentrations during rainfall recharge (Green et al., 2023). The results observed in this study highlight the temporal variation in biogeochemical processing rates in response to forestry harvesting and subsequent nutrient losses driven by hydrological conditions.

The recent application of UV nitrate sensors to in-situ groundwater poses challenges when comparing results to other studies with lower frequency measurements. At the study site, there is unlikely to be significant denitrification due to the thin, low-carbon pallic soils and gravel aquifers (Wilson et al., 2020). Therefore, we expect nitrate-nitrogen concentrations to be higher than other studies reporting stream concentrations that have longer hydrological flow paths leading to increased denitrification potential and stream uptake (Table 1). Other studies measuring inorganic nitrogen concentrations in drainage water generally report higher concentrations (Table 1), but it is unclear whether peak concentrations were captured using lower temporal resolution techniques. This study highlights the benefits of high-frequency measurements in settings where hydrological events drive rapid changes in nitrate-nitrogen concentrations and groundwater levels.

### 5.2    Hysteresis analysis findings

We used a novel approach of comparing high-frequency groundwater levels and nitrate-nitrogen concentrations to understand event scale responses to recharge. Hysteresis events in 2021 (Fig. 8) generally showed initial rapid increases in concentration (clockwise loop) indicating local leaching sources, that were rapidly mobilised. While in 2020, groundwater was less hydrologically connected to recharge, resulting in more complex hysteresis loops as a result of slower drainage pathways (Fig. 7). Across both years, the initial winter recharge event had a higher average area (0.65) compared to subsequent events (0.35). This finding indicates that stores of nitrate-nitrogen were initially rapidly mobilised, but subsequent events resulted in diluted or depleted stores of nitrate-nitrogen.

We observed nitrate-nitrogen concentration increases shifting from surface-based recharge to groundwater levels mobilising vadose zone storage. This was evident in event f (Fig. 8) at the end of winter in 2021 as there was a linear relationship between increasing groundwater levels and nitrate-nitrogen concentrations. We think this occurred due to successive antecedent rainfall recharge events over a short period of time flushing and exhausting soil storage. Mobilisation then occurred in the vadose zone as groundwater levels rose, intercepting slower drainage or stored pore water with higher nitrate-nitrogen concentrations from previous events (Ascott et al., 2017; Burbery et al., 2021). Several other hysteresis curves after the June 2021 recharge event (near the end of winter) also exhibited limited hysteresis. This indicates that the dual components of soil storage and vadose zone storage were mobilised during these events.

In concentration-discharge studies the input of near stream groundwater is recognised (Knapp et al., 2020; Winter et al., 2021; Gelmini et al., 2022) but the timing and variation of groundwater solute discharges is typically unknown. The novel hysteresis analysis shows that there was significant inter- and intra- annual variation between recharge events. These findings can be a valuable addition to improve hysteresis analysis in riverine environments, where groundwater is a significant contributor to baseflows. Ideally, future riverine hysteresis studies would consider groundwater hysteresis when conceptualising their study, so that high-frequency groundwater concentrations and water levels are available.

Groundwater concentration-groundwater level analysis can provide future riverine studies with more information on proximal (clockwise) or distal (anticlockwise) groundwater sources of nitrate-nitrogen. As well as the source of nitrate-nitrogen, the timing of greatest surface water-groundwater connection (highest groundwater level) compared to highest nitrate-nitrogen concentration could be investigated. This study highlights that the nitrate-nitrogen concentration in groundwater can be just as varied as stream concentrations at the event scale and is a valuable addition in understanding riverine hysteresis processes.

### 5.3 Nitrate-nitrogen leaching and export

High-frequency monitoring indicates there are distinct pulses of nitrate-nitrogen to the Hurunui River from forestry land use (Fig. 6). The distinct surface runoff and groundwater discharge pathways result in different timings of nitrate-nitrogen in the Hurunui River. The surface runoff component appears to occur simultaneously with high flow volumes in the Hurunui River. While the lag associated with the groundwater travel time indicates that the groundwater discharge generally occurs when there are base flow conditions in the Hurunui River (Fig. 6).

The hysteresis analysis and exported load estimates suggest that nitrate-nitrogen stores were small and quickly depleted during recharge events. The largest nitrate-nitrogen export observed in 2020 resulted from several factors: the shortest time period since the last forestry disturbance in 2018, preceding dry conditions (Winter et al., 2023), and repeated wetting and recharge events (Anaya et al., 2006). In contrast, subsequent years saw larger but less frequent rainfall events, leading to different export characteristics (Table 2). Annual load estimates were comparable to other studies estimating the nutrient loads from forestry land use, despite the high concentrations we observed (Di and Cameron, 2002; Gundersen et al., 2006; Davis, 2014). However, when analysing nutrient concentrations in the Hurunui River, the effects of forestry-related nutrient losses cannot be isolated from those of surrounding land uses.

The high-frequency UV nitrate sensor data and integration method was better at constraining the nitrate-nitrogen concentration and therefore the load. Using discrete annual and quarterly values did not accurately estimate the nitrate-nitrogen load depending on the timing of the sample (Table 3). The results from this study highlight the importance of choosing an appropriate monitoring frequency and statistical description, that can describe pulses of nitrate-nitrogen. There are still some weaknesses in this method of estimating the nitrate-nitrogen load as the volume of groundwater that was discharging to surface water was uncertain. Additionally, the variation in nitrate-nitrogen concentration across the aquifer thickness was also not measured. Multiple UV nitrate sensors at different depths or multiple pumps with a flow cell would improve this information.

The data from the high-frequency UV nitrate sensor provides important information on the timing of leaching and improvements in constraining nitrate-nitrogen loads. For regulators that aim to improve water quality, these

findings provide more understanding on the effects of changing land use and allow more targeted approaches to policy as there is greater information available on the timing and causes of nutrient losses. The strong hydrological controls on the leaching of nitrate-nitrogen can be applied to future climate change projections. With increased periods of droughts and intensity of storm events, we can expect to see more pulses of nitrate-nitrogen in the future, which may exacerbate existing water quality stressors on water resources (Srinivasan et al., 2021).

## 6    Conclusions

This study presents four years of high-frequency nitrate-nitrogen concentrations and groundwater levels within a forestry catchment. As one of the first studies to use an in-situ UV nitrate sensor in this setting, we observed distinct pulses of nitrate-nitrogen in winter months. These pulses were exacerbated by prior, low recharge conditions and forestry harvesting during the study, both of which increased the available nitrate-nitrogen in the catchment. Our study provides valuable insights into nitrate-nitrogen leaching dynamics at the event scale through the novel approach of using high-frequency nitrate-nitrogen and groundwater level measurements in hysteresis analysis. We found that the mechanism of increases in nitrate-nitrogen changed over winter from soil leaching, to vadose zone stores being mobilised from rising groundwater levels after successive recharge events. In this dynamic groundwater environment, we found that the method of integrating the high-frequency UV nitrate sensor data was better at constraining the flux of nitrate-nitrogen to the Hurunui River. These findings have strong implications for land management and future climate change projections. Dry periods with low recharge and storm events were found to be strong hydrological controls on concentrations of nitrate-nitrogen observed in groundwater. The frequency of these events is expected to increase in the future, indicating the potential for the increase of pulses of inorganic nitrogen.

*Data availability.* The data is available on request from Environment Canterbury Regional Council.

*Supplementary information.*

*Author contributions.* BW performed the analyses, created the figures and prepared the manuscript. TJ gathered
the data. SM contributed to editing and review.

*Competing interests.* The authors declare that they have no competing interests.

*Review statement.*

*Financial support.* The authors did not receive funding for the writing and publishing of this article

*Acknowledgements.* We thank the groundwater and surface water field teams at Environment Canterbury for
collecting the data at the study location. We also thank the reviewers of this manuscript who provided helpful
comments and discussion on the findings and analysis.

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
