# Peer review of "Nitrate-nitrogen dynamics in response to forestry harvesting and climate variability: Four years of UV nitrate sensor data in a shallow, gravel aquifer"

_EGUsphere, 2024_

## Referee Comment (RC2)

L10: 'discrete' – I would suggest to use something like 'low frequency', because even *in situ* sensors provide discrete measurements.

L12: 'continuous' – would not use this as adjective, as the sensor is not continuous. You could emphasise the high frequency data collection with 'measuring at xxx minute interval'. Or call them 'high-frequency sensors', as you do in L99 (and please use the same term consistently).

L16 and whole document: are concentration in mg NO3 per L or mg N per L? Please clarify in the manuscript.

L19-20: 'larger average area' – What are the implications of a larger or smaller area (or more or less hysteresis)? What can we learn from this information? Might be too complicated to include in the abstract and needs to be explained properly in the main text.

L44: 'forestry ecosystems' – Does the description of processes in the following sentences refer to forestry ecosystems only (which, I presume, are plantation or intensively managed forests only?) or all forest ecosystems? If it applies to all forests, I would use the term 'forest ecosystems' instead.

Table 1: Include the abbreviation 'NO3-N' in the caption behind 'nitrate-nitrogen'. Interesting additional information for the table would be an indication of the time period over which each study was conducted and potentially even the method used to obtain the nitrate data.

L100-103: I presume that you mean that high frequency monitoring could be done in multiple places to understand the high variability in inorganic N movement? Or do you mean that one high-frequency sensor can help to better understand the spatial variability as well? It might be good to clarify this in the sentence.

L133-140: While the objective makes sense for a case study perspective, the 'bigger picture' is missing. Why is this study relevant for others than those working in the case study location? What can be learned from it, applied elsewhere?

Section 2: You could include some more background information on the climate in the study area, particularly rainfall amounts and seasonality (if applicable) and evapotranspiration.

L177: 'and telemetry' – Is this really described in this section? I could not identify it.

L207-208: 'were adjusted to concurrent validation measurement' – This sound to me like you calibrated the sensor using the validation sample measurements (and the annual/quarterly measurements?). Perhaps use the word 'calibration' instead of your current formulation?

Section 3.1: Did you do any data post-processing or quality control (other than comparison to grab samples) to the time series data? E.g. outlier detection, gap-filling, etc. If so, please describe.

Section 3.2: It is not clear on which information the choice of parameter values for soil water capacity, evaporation reduction function, crop factor and drainage threshold were based, nor where the data for precipitation (from the rain gauge next to the well?) and Penman PET were obtained. Please add this information.

Section 3.3 and 3.4: It is not entirely clear to me how these data will be used (i.e. related to the groundwater N data) based on the description. It would be helpful if this could be highlighted in a sentence in each of the sections.

L225-226: Was the sensor also adjusted using these monthly grab samples, like with the sensor installed in the well?

L235: This is probably the question you do not want to get, but how was the 1 mg/L threshold determined?

L235-237: What was the point of splitting up the time period? Was the rate of change determined for each of the five sections?

L241-243: Could you explain the relevance of the different metrics? What do they indicate?

L245-246: Since you mention how many events were identified in 2022 and 2023, it would be interesting to report in this paragraph as well, how many events were identified for 2020 and 2021.

L252: 'bore' – Should be 'borehole'?

L252-257: It is not entirely clear what you are describing here. I think you are not calculating the nitrate load yet, since you are only using the concentration data. The load calculation itself seems to be described in L265-267, where you multiply the concentration with the volume of groundwater discharged from the aquifer. Or how does the area under the curve represent the load?

L294-295: 'In November 2020, [...] flow and rainfall' – This comparison is a bit odd. What is the point of comparing the highest concentration in Nov. 2020 with the highest concentration in Feb. 2022? Unless you want to point something out (which you would have to specify), I would take it out or reformulate in a way that it does not become an odd comparison.

L298-299: 'Data was missing [...] and interference effects.' – This information could be included in the methods, including information on how much data (%) is missing. Were there no data gaps at all for the sensor deployed in the groundwater? Please specifically mention this (or the percentage of data missing) in the methods as well. I was wondering whether the sudden drops in nitrogen concentrations in the Hurunui River were also related to maintenance (e.g. cleaning), but since some of the increases are also reflected in the monthly nitrate data, I might be mistaken.

L306: I would refer to Fig. 5 in L307, following '[...] low rainfall recharge in Canterbury.'

L307: Where is Balmoral? How does this site relate to the sites included in Fig. 5?

Section 4.2: These sections seem a bit isolated from the rest of the manuscript. They could be embedded better by directly linking it to the groundwater nitrogen data (although that might require merging of result and discussion?) or they could be placed somewhere else than between the initial description of the groundwater nitrogen data and the hysteresis analysis, which is based on the same data.

Section 4.3/Fig. 7-8: Are these six the only hysteresis curves you analysed (i.e. the only identified events)? This could be clarified by including in the text how many events were identified and analysed.

L348-356: A lot of this information is difficult to interpret, because it is not clear what the relevance of the average area is and incomplete formulation. For example, in the methods you state 'the proportion of time into the event that the peak groundwater level and nitrate-nitrogen concentration [occurs]', which is a lot easier to understand than 'time to reach peak groundwater level and peak nitrate-nitrogen concentration' expressed in a percentage. Or 'The residual analysis indicated a return to pre-event conditions...', was this done for all events individually?

Fig. 9: Please include the unit (%) for the first two graphs and add 'of hysteresis curve' to the label of the y-axis of the third graph. Also include on how many events these boxplots are based.

L382-383, Table 3: If I understand correctly, these loads are based on the groundwater nitrate data and groundwater validation samples. Did you also compare these loads with export values calculated from the river nitrate data (at least for those periods for which there is data available for both sensors)?

L388-395: It would be interesting to see a time series plot of the load data to better visualise the pulses and their timing.

Section 5: Since the discussion is relatively short, I would recommend integrating it with the results. This avoids duplication of information and might make the interpretation of some of the data also more straightforward (see previous comment about interpretation of hysteresis metrics).

L404-407: Can you really fully attribute these differences to the harvesting? You point out in Section 4.2.1 that this period is also characterised by a change in weather (not climate!) from dry conditions to wetter conditions.

L426-428: 'Over consecutive recharge events [...] after successive recharge events.' – either remove the end or the beginning of the sentence to avoid duplication.

L437-438: I see how this approach could complement the more 'traditional' analysis of concentration-discharge relationships or hysteresis analysis, but I think the formulation that the approach you used can be applied to improve hysteresis analysis in streams and riverine environments in incorrect. It can only be applied in cases whereby groundwater data (concentrations and levels) is available.

---

## Author Response (AR1)

**Response to review**

**Reviewer #1**

**General comments**

1. Clear research objectives are addressed, but there is no leading research question(s) presented.

Thank you for this comment, we agree that more direction on the purpose of the research is needed. We have added more detail on the leading research question from lines 155-160 in the revised manuscript.

2. The introduction provides a good overview of the problems and methods but could benefit from highlighting the consequences of disturbance on forestry nutrient leaching, e.g., concluding or comparing the difference in nitrate-nitrogen concentration before and after ecosystem change for the six disturbed forests listed in Table 1 (if data is available in your references).

To highlight the changes in nutrient leaching after disturbance, we have added an explanation that the higher nitrate-nitrogen concentration is from disturbance in Table 1. We have also added a comment on where the highest nutrient leaching has been observed (lines 75-80).

3. Although only the hysteresis analysis for winter months is shown in your supplement, do you have any findings or insights for the correlation between groundwater level and nitrate-nitrogen leaching during droughts?

The supplement shows all hysteresis events from 2020 through to the end of 2021 during summer and winter. However, the majority occurred during winter because of the timing of rainfall recharge and nitrate-nitrogen availability. A comparison is difficult because of the low sample size in summer. We comment on the difference in hysteresis curves between wet and dry years.

**Specific comments**

1. Line 160: Add the coordinates to the monitoring site.

We have added the coordinates of the monitoring bore.

2. Line 170: How do you account for the effects of nitrogen leaching from irrigated beef on the nitrate-nitrogen concentration in the Hurunui River?

We explain in the discussion that the leaching from different land uses cannot be separated in the recorded nitrate-nitrogen concentration from the Hurunui River. We comment on the likely effect that forestry is having on Hurunui River compared to other land uses given the load calculated in this study and the findings from other studies.

3. Line 173, Figure 1: Is it possible to show the location of the lysimeter on the map?

We have added the lysimeter and automatic weather station to Figure 1.

4. Line 179: How does the path length setting of the nitrate sensor affect its measurement accuracy? What might be the causes for the overestimation of high nitrate-nitrogen concentrations?

We have added more detail to the manuscript on the accuracy of the UV nitrate sensor. We talk about the accuracy of the path length used in the study and how the accuracy of the UV nitrate sensor cannot be optimised for all conditions, which explains how it can overestimate high nitrate-nitrogen concentrations.

5. Line 190: What's the precision of the TriOS NICO sensor during the study period?

We cannot reliably report the precision of the UV nitrate sensor because the standards do not have constituents that cause interference. Therefore, the in-situ precision and standard precision are different.

6. Line 207-208: How did you adjust/calibrate the UV nitrate sensors?

We have rewritten this section so that it is clear that we are commenting on data post-processing methods (lines 246-250).

7. Line 213-215: Which land surface recharge model did you use? Is it a 1-D model? What is its principle for calculating land surface recharge? How did you define the values of parameters? From your reference, I learnt that the model is named the soil-water balance model (GDA-LSR). It's better to introduce a bit of the model and justify model performance because rainfall recharge is one of your main studied variables.

We have added details to this section on the variables used in the land surface recharge model (Section 3.2). We explain that rainfall recharge occurs when rainfall is above the soil water storage capacity. Evaporation, plant uptake and runoff are factors that reduce soil water storage.

8. Line 225: At what depth was the UV nitrate sensor installed in the Hurunui River?

The depth varied based on Hurunui River flows during summer and winter.

9. Line 227: What is the difference between "rainfall recharge" here and that calculated from the land surface model in Line 212?

We have clarified that this section (3.4) is an indicator of climate variation, of which rainfall recharge is a component.

10. Line 241: What's the HARP (full name?) algorithm, and what do values represent in Line 348, e.g., 0.65, 0.35 and 50% and 55%?

We have added details on the HARP name, analysis methods and results in section 3.5 (lines 305 to 316).

11. Line 259: Does "the volume of water moving in a horizontal direction from the aquifer to the Hurunui Rive" refer to the same as groundwater discharge (qi) in Eq. (1)? Can you introduce the equation used for calculating qi?

We have introduced the equation, and its components used to calculate groundwater discharge in section 3.6.

12. Line 275: Add units to each variable used in the equations and provide clear definitions.

We have added units to variables in equation 2 and improved the explanation.

13. Line 291: Why did the groundwater level remain low and stable from 2020 to June 2021?

We have rewritten this section (4.1.2) to be clearer on what we are showing in the figure.

14. Line 306: Include a statement clarifying the two distinct climate types and how they influenced the rates of nitrate-nitrogen leaching.

The discussion comments on the two distinct climate types during the study period and how they influenced nitrate-nitrogen leaching (Section 5.1).

15. Line 355: What is the residual analysis for and what is meant by "not changed system state" in Fig. 4?

The residual analysis is explained in section 3.5. We also explain why we did not use this analysis (lines 312 to 315).

16. Line 383: How did you end up with the conclusion of "The annual and quarterly measurements over or underestimated the load depending on whether they captured the winter pulses of nitrate-nitrogen" from the data in Table 3?

We have changed the wording of this statement to be clearer and referenced figure 2.

17. Line 385: "As a sensitivity analysis" for groundwater discharge calculation?

We have moved this comment to the methods section and provided an explanation and references on the methodology used (lines 360 to 365).

18. Line 402: "We infer that the reduced nutrient demand, increased rainfall and favourable soil water balance conditions induced higher nutrient losses from the forest soils during winter", where the "favourable soil water balance conditions" is a general indication, could you specify what kind of soil water balance conditions based on your findings in this study?

We have simplified this statement to say that the saturated soil conditions were one aspect that induced higher nutrient losses during winter.

**Technical comment**

1. Line 128: Add a punctuation mark before "Conversely".

2. Line 168: Add a comma before "sections".

3. Line 192: Add a punctuation mark after "(Fig. 2)".

4. Line 215: Add the unit to "soil water capacity (86.0 mm)".

5. Line 252: Add a comma after the "bore".

6. Line 262: Add a comma after the "…bgl)".

We have made these changes.

7. Line 301: Just some suggestions for better visualisation of Figure 4, (1) Between two consecutive years on x-axis, add scale bars with monthly timestep will help readers to identify the winter and summer months easier; (2) Label each plot with (a)-(g) for future reference; (3) Combine the last two graphs, e.g., plotting the point series data as the secondary y-axis with different colour scheme on the continuous dataset, to compare lab measurements and UV sensor data while saving space.

We have labelled each section of the graph from a-g so that it is easy to refer to. We have also added quarterly scale bar intervals between years for readability. We decided not to combine the high-frequency and discrete Hurunui River datasets because they are different data types, and we want to be able to easily refer to them.

8. Line 311: Figure 5, the well count for red groups seems to be in the reserve order, then it should range from [0%, 100%] from the top to the bottom at the y-axis on the right hand side.

The well counts add to 100% so we don't think there is a need for two y-axes. We have increased the interval of the y-axis so that it is easier to read.

Reviewer #2

(1) Some details are missing in the methods section (e.g. climate of study area, information on post-processing, more details on the modelling). Details are provided in the detailed comments attached. I also think that the load calculation explanation could use some clarification and it should be better explained how the Hurunui River data is being used.

We have updated the methods section to have more detail on the climate of the study area, the methods used in post-processing data and a clearer description of the land surface recharge model. We also explain why we have included the Hurunui River data.

(2) From only reading the manuscript, it is not clear how many hysteresis events were finally analysed. This could be clarified in the text, as well as in the box plots with the hysteresis metrics.

We clarified in the manuscript that 25 events were analysed, with 15 events occurring in 2020 and 10 events in 2021.

(3) The manuscript could benefit from an integration of the results and discussion, as the discussion section is relatively short and some of the results are hard to interpret based on the current formulation. It might also make the flow more logical, as section 4.2 currently breaks the flow from the description of the time series nitrogen data to the hysteresis analysis which uses the same data.

We have changed the results section so that the order is easier to follow, and we have added more detail to the discussion section. We think the current iteration of the manuscript clearly communicates the results and lead to the discussion.

(4) I am slightly sceptical about the use of a change in nitrate concentration to identify events, especially since the threshold of 1 mg/L is not necessarily justified. Similar to other event-based analyses, I would rather let it be guided by the occurrence of rainfall recharge. I think the authors need to better justify their choice of event identification.

We have added detail in the methods section to explain why we chose a threshold of 1 mg L$^{-1}$ (accuracy of nitrate-nitrogen and groundwater level measurements as well as variation of baseline conditions). We have not used rainfall recharge to identify events because we identified increases in nitrate-nitrogen without immediately prior rainfall recharge.

L10: 'discrete' – I would suggest to use something like 'low frequency', because even in situ sensors provide discrete measurements.

We think discrete is a term that is widely used to indicate grab samples or single samples and accurately describes the sample type.

L12: 'continuous' – would not use this as adjective, as the sensor is not continuous. You could emphasise the high frequency data collection with 'measuring at xxx minute interval'. Or call them 'high-frequency sensors', as you do in L99 (and please use the same term consistently).

We have changed the description of the UV nitrate sensor measurements to high-frequency. We have also checked the manuscript to ensure that the adjectives describing the UV nitrate sensor measurements are consistent.

L16 and whole document: are concentration in mg NO3 per L or mg N per L? Please clarify in the manuscript.

We clarify at the beginning of the methods section that the results are reported as milligrams of nitrate-nitrogen per litre.

L19-20: 'larger average area' – What are the implications of a larger or smaller area (or more or less hysteresis)? What can we learn from this information? Might be too complicated to include in the abstract and needs to be explained properly in the main text.

We have added a new paragraph that describes what the HARP analysis in terms of physical processes in the catchment during rainfall recharge (lines 317 to 327).

L44: 'forestry ecosystems' – Does the description of processes in the following sentences refer to forestry ecosystems only (which, I presume, are plantation or intensively managed forests only?) or all forest ecosystems? If it applies to all forests, I would use the term 'forest ecosystems' instead.

We have changed the description of generalised processes that occur to refer to forests. As our case study and references describe forestry harvesting, we have left that term where we think it is more applicable.

Table 1: Include the abbreviation 'NO3-N' in the caption behind 'nitrate-nitrogen'. Interesting additional information for the table would be an indication of the time period over which each study was conducted and potentially even the method used to obtain the nitrate data.

We have added the abbreviation. Additionally, we have added the duration and approximate sample frequency of the studies to the table.

L100-103: I presume that you mean that high frequency monitoring could be done in multiple places to understand the high variability in inorganic N movement? Or do you mean that one highfrequency sensor can help to better understand the spatial variability as well? It might be good to clarify this in the sentence.

We have clarified this sentence so that it easier to understand that high frequency monitoring can be useful in multiple settings.

L133-140: While the objective makes sense for a case study perspective, the 'bigger picture' is missing. Why is this study relevant for others than those working in the case study location? What can be learned from it, applied elsewhere?

We have added more detail on the leading research question and the objectives of the study.

Section 2: You could include some more background information on the climate in the study area, particularly rainfall amounts and seasonality (if applicable) and evapotranspiration.

We have included a description of the climate in the site description.

L177: 'and telemetry' – Is this really described in this section? I could not identify it.

We have removed telemetry from the heading.

L207-208: 'were adjusted to concurrent validation measurement' – This sound to me like you calibrated the sensor using the validation sample measurements (and the annual/quarterly measurements?). Perhaps use the word 'calibration' instead of your current formulation?

We have removed this sentence and added a section that describes the post-processing methods (lines 246 to 251).

Section 3.1: Did you do any data post-processing or quality control (other than comparison to grab samples) to the time series data? E.g. outlier detection, gap-filling, etc. If so, please describe.

We describe the post-processing methods from lines 246-251.

Section 3.2: It is not clear on which information the choice of parameter values for soil water capacity, evaporation reduction function, crop factor and drainage threshold were based, nor where the data for precipitation (from the rain gauge next to the well?) and Penman PET were obtained. Please add this information.

We have added more details on the use of parameters in the land surface recharge model. Please see section 3.2.

Section 3.3 and 3.4: It is not entirely clear to me how these data will be used (i.e. related to the groundwater N data) based on the description. It would be helpful if this could be highlighted in a sentence in each of the sections.

We have changed these sections and clarified the purpose of including them with an initial statement.

L225-226: Was the sensor also adjusted using these monthly grab samples, like with the sensor installed in the well?

We have added more details on the UV nitrate sensor in the Hurunui River.

L235: This is probably the question you do not want to get, but how was the 1 mg/L threshold determined?

We have added detail in the methods section to explain why we chose a threshold of 1 mg L$^{-1}$ (accuracy of nitrate-nitrogen and groundwater level measurements as well as variation of baseline conditions). We have not used rainfall recharge to identify events because we identified increases in nitrate-nitrogen and groundwater levels without immediately prior rainfall recharge.

L235-237: What was the point of splitting up the time period? Was the rate of change determined for each of the five sections?

We visually show the rate of change in the graphs by using different colours. This also helps with identifying the start and end of the event.

L241-243: Could you explain the relevance of the different metrics? What do they indicate?

We have added a new paragraph that describes what the HARP analysis in terms of physical processes in the catchment during rainfall recharge (lines 317 to 327).

L245-246: Since you mention how many events were identified in 2022 and 2023, it would be interesting to report in this paragraph as well, how many events were identified for 2020 and 2021.

We have added that there were 15 events in 2020 and 10 events in 2021

L252: 'bore' – Should be 'borehole'?

It is commonly referred to as a monitoring bore in New Zealand. We have changed it to borehole for a wider audience.

L252-257: It is not entirely clear what you are describing here. I think you are not calculating the nitrate load yet, since you are only using the concentration data. The load calculation itself seems to be described in L265-267, where you multiply the concentration with the volume of groundwater discharged from the aquifer. Or how does the area under the curve represent the load?

Thank you for pointing this out. We have clarified that we are not referring to the load in this section.

L294-295: 'In November 2020, […] flow and rainfall' – This comparison is a bit odd. What is the point of comparing the highest concentration in Nov. 2020 with the highest concentration in Feb. 2022? Unless you want to point something out (which you would have to specify), I would take it out or reformulate in a way that it does not become an odd comparison.

We have changed this sentence to highlight that we are comparing summer events with and without prior forestry harvesting.

L298-299: 'Data was missing […] and interference effects.' – This information could be included in the methods, including information on how much data (%) is missing. Were there no data gaps at all for the sensor deployed in the groundwater? Please specifically mention this (or the percentage of data missing) in the methods as well. I was wondering whether the sudden drops in nitrogen concentrations in the Hurunui River were also related to maintenance (e.g. cleaning), but since some of the increases are also reflected in the monthly nitrate data, I might be mistaken.

We have added a paragraph in the methods section (3.1) that describes what we did during post-processing of the data. We also further describe the methods and data of the Hurunui River UV nitrate sensor.

L306: I would refer to Fig. 5 in L307, following '[…] low rainfall recharge in Canterbury.'

We now refer to Figure 5.

L307: Where is Balmoral? How does this site relate to the sites included in Fig. 5?

We have added a map to this figure to show where the regional groundwater levels were measured in relation to the study site. This provides context on where the study site is in relation to the Canterbury region.

Section 4.2: These sections seem a bit isolated from the rest of the manuscript. They could be embedded better by directly linking it to the groundwater nitrogen data (although that might require merging of result and discussion?) or they could be placed somewhere else than between the initial description of the groundwater nitrogen data and the hysteresis analysis, which is based on the same data.

We have reordered this section to have the results that show why there might be high nitrate-nitrogen concentrations before presenting the timeseries graph. We think this improves the readability of the results section.

Section 4.3/Fig. 7-8: Are these six the only hysteresis curves you analysed (i.e. the only identified events)? This could be clarified by including in the text how many events were identified and analysed.

We clarify that there were 15 events analysed in 2020 and 10 in 2021.

L348-356: A lot of this information is difficult to interpret, because it is not clear what the relevance of the average area is and incomplete formulation. For example, in the methods you state 'the proportion of time into the event that the peak groundwater level and nitrate-nitrogen concentration [occurs]', which is a lot easier to understand than 'time to reach peak groundwater level and peak nitrate-nitrogen concentration' expressed in a percentage. Or 'The residual analysis indicated a return to pre-event conditions…', was this done for all events individually?

In the revised manuscript, we provide a description of what the HARP analysis findings mean in the methods section. We have also changed the language so that it is consistent and clear.

Fig. 9: Please include the unit (%) for the first two graphs and add 'of hysteresis curve' to the label of the y-axis of the third graph. Also include on how many events these boxplots are based.

We have added these details to the graphs and captions.

L382-383, Table 3: If I understand correctly, these loads are based on the groundwater nitrate data and groundwater validation samples. Did you also compare these loads with export values calculated from the river nitrate data (at least for those periods for which there is data available for both sensors)?

We have not presented the load from the Hurunui River data. The explanation required would be considerable and we are not sure how to fit this into the current version of the manuscript. We instead comment that the exported load from groundwater and forestry land use is a small component of the Hurunui River concentration, compared to the wider land use (dairy and beef farming).

L388-395: It would be interesting to see a time series plot of the load data to better visualise the pulses and their timing.

The load data is very similar to the concentration data. However, because of the lag time associated with groundwater flow we are not sure how effective a graph of the load would be at showing the timing of the pulses.

Section 5: Since the discussion is relatively short, I would recommend integrating it with the results. This avoids duplication of information and might make the interpretation of some of the data also more straightforward (see previous comment about interpretation of hysteresis metrics).

We think the revised manuscript clearly presents the results and discussion in separate sections.

L404-407: Can you really fully attribute these differences to the harvesting? You point out in Section 4.2.1 that this period is also characterised by a change in weather (not climate!) from dry conditions to wetter conditions.

We have included rainfall volumes in the comparison to be clear that it is not just harvesting.

L426-428: 'Over consecutive recharge events […] after successive recharge events.' – either remove the end or the beginning of the sentence to avoid duplication.

We have rewritten this sentence to avoid duplication.

L437-438: I see how this approach could complement the more 'traditional' analysis of concentration-discharge relationships or hysteresis analysis, but I think the formulation that the approach you used can be applied to improve hysteresis analysis in streams and riverine environments in incorrect. It can only be applied in cases whereby groundwater data (concentrations and levels) is available

We have changed this section to be clearer. We think groundwater hysteresis could be valuable for future studies if it is recognised during study conceptualisation.

---

## Referee Report (RR1)

The manuscript titled "*Nitrate-nitrogen dynamics in response to forestry harvesting and climate variability: Four years of UV nitrate sensor data in a shallow, gravel aquifer*" explores nitrate-nitrogen dynamics in groundwater and riverine systems influenced by forestry harvesting and climate variability. The revised manuscript clarifies the methodology very well and sharpens the result analysis and discussion. However, the **overall method introduction can be streamlined** for better readability, such as the hysteresis approach and content in Line 250-258:

"*A 1-D land surface recharge model was calibrated using nearby lysimeter data to determine when rainfall recharge occurred. The parameters used in the land surface recharge model  included precipitation and Penman PET data sourced from a nearby weather station (Fig. 1). Ssoil water capacity (86.0 mm),  evaporation reduction function (10.0) and drainage threshold (50.0) were derived from the nearby lysimeter (Fig. 1), while the crop factor (0.77) was sourced from Allen et al (1998)  (Bidwell and Burbery, 2011).  In this model, rainfall recharge occurs when rainfall is above the soil water storage capacity. Evaporation, plant uptake and runoff are factors that reduce soil water storage.*"

**Some minor revisions could be considered before final publication:**

1.  Line 327: Add "(1)" to the right end of this equation

2.  Line 354: Add "(2)" to the right end of this equation

3.  Line 347: "*This resulted in a 25 percent increase or decrease in exported nitrate-nitrogen (Table 2)*". Referring to Table 2 improves readability by providing clarity on source of ($\pm$ values).

4.  Fig. 5: "*The map of the South Island of New Zealand shows the Canterbury region where the groundwater level measurements were taken and the area of the study site.*"

    You added this map in response to one of the reviewers' comments: "*L307: Where is Balmoral? How does this site relate to the sites included in Fig. 5?*" Your response was: "*We have added a map to this figure to show where the regional groundwater levels were measured in relation to the study site. This provides context on where the study site is in relation to the Canterbury region.*"

    However, there is no clear indication or relationship to Balmoral in the added map. Furthermore, this map appears similar to the one in the top-right corner of Fig. 1. Consider explicitly pointing out Balmoral on this map and placing it in the top-right corner of Fig. 1, instead of introducing the study area in the Results section.

5.  Line 468: "*There were higher intensity rainfall recharge events in 2021 and more rainfall recharge (265 mm) but the nitrate-nitrogen exported was slightly less than 2020 at 6.91 t yr$^{-1}$.*"

Could the causes of this observation be further elaborated in Section 5.3? Based on your previous statements, my understanding is that the lower nitrate-nitrogen export in 2021, despite higher-intensity rainfall recharge events, is due to the depletion of nitrate-nitrogen stores after the initial recharge events (i.e., larger hysteresis areas and rapid mobilization of nitrate-nitrogen). In contrast, in 2020, these stores were less depleted, resulting in higher cumulative export over the year. Please correct me if my interpretation is incorrect, and consider discussing this result in Section 5.3.

6. Line 515-523: this paragraph is not discussing the drivers of nitrate-nitrogen dynamics but comparing the measured/calculated nitrate-nitrogen concentrations with other literature (Table 1). For better coherence, consider merging this content with the paragraph at Line 565 to consolidate your discussion on the benefit of using high-frequency UV nitrate sensor data and integration method.

---

## Author Response (AR2)

**Author's response**

**Report #1 comments**

Thank you for your review and comments. We agree that these changes will improve the manuscript.

Section 4.1: First paragraph reads more as methods. No results are presented until 4.1.1.

We have rewritten the introduction to the results section to cover what we found and the importance of land use change and hydrological conditions.

Fig. 5b could go into methods (map) where you describe the use of groundwater well levels.

Added another inset map to Fig. 1, that shows the location of the study site (Balmoral) and the Canterbury region. This figure is referred to in the figure description of Fig. 5.

L568–569: "Over consecutive [...] vadose zone storage." Sentence is somewhat duplicated.

Agree. Have simplified and removed duplication in these statements.

**Report #2 comments**

Thank you for your review and comments. We appreciate your detailed comments and suggested changes. We agree that these changes will improve the manuscript.

The overall method introduction can be streamlined for better readability, such as the hysteresis approach and content in Line 250-258.

The methodology of calculating rainfall recharge has been simplified as suggested.

We have also removed duplicate statements in the hysteresis analysis methods.

Line 327: Add "(1)" to the right end of this equation

We have added an equation number.

Line 354: Add "(2)" to the right end of this equation

We have added an equation number.

Line 347: "This resulted in a 25 percent increase or decrease in exported nitrate-nitrogen (Table 2)". Referring to Table 2 improves readability by providing clarity on source of (± values).

Agree, have added a reference to Table 2 in this sentence.

(Fig. 5): There is no clear indication or relationship to Balmoral in the added map. Furthermore, this map appears similar to the one in the top-right corner of Fig. 1. Consider explicitly pointing out Balmoral on this map and placing it in the top-right corner of Fig. 1, instead of introducing the study area in the Results section.

We have added another inset map to Fig. 1 to show where the Balmoral study site and Canterbury region are located more clearly and earlier in the text.

We refer to Fig. 1 in the figure description of Fig. 5 for clarity.

Line 468: "There were higher intensity rainfall recharge events in 2021 and more rainfall recharge (265 mm) but the nitrate-nitrogen exported was slightly less than 2020 at 6.91 t yr-1 ." Could the causes of this observation be further elaborated in Section 5.3? Based on your previous statements, my understanding is that the lower nitrate-nitrogen export in 2021, despite higher-intensity rainfall recharge events, is due to the depletion of nitrate-nitrogen stores after the initial recharge events (i.e., larger hysteresis areas and rapid mobilization of nitrate-nitrogen). In contrast, in 2020, these stores were less depleted, resulting in higher cumulative export over the year. Please correct me if my interpretation is incorrect and consider discussing this result in Section 5.3.

We provide a more detailed explanation on why the exported loads vary between the initial years in section 5.3.

Line 515-523: this paragraph is not discussing the drivers of nitrate-nitrogen dynamics but comparing the measured/calculated nitrate-nitrogen concentrations with other literature (Table 1). For better coherence, consider merging this content with the paragraph at Line 565 to consolidate your discussion on the benefit of using high-frequency UV nitrate sensor data and integration method.

We have moved the second part of this paragraph that deals with load estimates to section 5.3. We have kept the initial sentences in this paragraph and expanded on the drivers of changing biogeochemical conditions in relation to observed nitrate-nitrogen concentrations.